# Differentiable Rendering with Perturbed Optimizers

**Quentin Le Lidec, Ivan Laptev, Cordelia Schmid and Justin Carpentier**

Inria - Département d'Informatique de l'École normale supérieure, PSL Research Un iversity

`{quentin.le-lidec,ivan.laptev,cordelia.schmid,justin.carpentier}@inria.fr`

## Abstract

Reasoning about 3D scenes from their 2D image projections is one of the core problems in computer vision. Solutions to this inverse and ill-posed problem typically involve a search for models that best explain observed image data. Notably, images depend both on the properties of observed scenes and on the process of image formation. Hence, if optimization techniques should be used to explain images, it is crucial to design differentiable functions for the projection of 3D scenes into images, also known as differentiable rendering. Previous approaches to differentiable rendering typically replace non-differentiable operations by smooth approximations, impacting the subsequent 3D estimation. In this paper, we take a more general approach and study differentiable renderers through the prism of randomized optimization and the related notion of perturbed optimizers. In particular, our work highlights the link between some well-known differentiable renderer formulations and randomly smoothed optimizers, and introduces *differentiable perturbed renderers*. We also propose a variance reduction mechanism to alleviate the computational burden inherent to perturbed optimizers and introduce an adaptive scheme to automatically adjust the smoothing parameters of the rendering process. We apply our method to 3D scene reconstruction and demonstrate its advantages on the tasks of 6D pose estimation and 3D mesh reconstruction. By providing informative gradients that can be used as a strong supervisory signal, we demonstrate the benefits of perturbed renderers to obtain more accurate solutions when compared to the state-of-the-art alternatives using smooth gradient approximations.

## 1 Introduction

Many common tasks in computer vision such as 3D shape modelling [5, 15, 34, 36, 37] or 6D pose estimation [18, 22, 25, 30, 32] aim at inferring 3D information directly from 2D images. Most of the recent approaches rely on (deep) neural networks and thus require large training datasets with 3D shapes along with well-chosen priors on these shapes. Render & compare methods [18, 22] circumvent the non-differentiability of the rendering process by learning gradients steps from large datasets. Using a more structured strategy would allow to alleviate the need for such a strong supervision.Using a more structured strategy would allow to alleviate the need for such a strong supervision. In this respect, differentiable rendering intends to model the effective image generation process to compute a gradient related to the task to solve. This approach has the benefit of containing the prior knowledge of the rendering process while being interpretable. This makes it possible to provide a supervision for neural networks in a weakly-supervised manner [10, 16, 23]. The main challenge of differentiable rendering lies in the non-smoothness of the classical rendering process. In a renderer, both the rasterization steps, which consist in evaluating the pixel values by discretizing the 2D projected color-maps, and the aggregation steps, which merge the color-maps of several objects along the depth dimension by using a Z-buffering operation, are non-differentiable operations (Fig. 1). Intuitively, these steps imply discontinuities in the final rendered image with respect to the 3D positions of the scene objects. For example, if an object moves on a plane parallel to the

35th Conference on Neural Information Processing Systems (NeurIPS 2021).

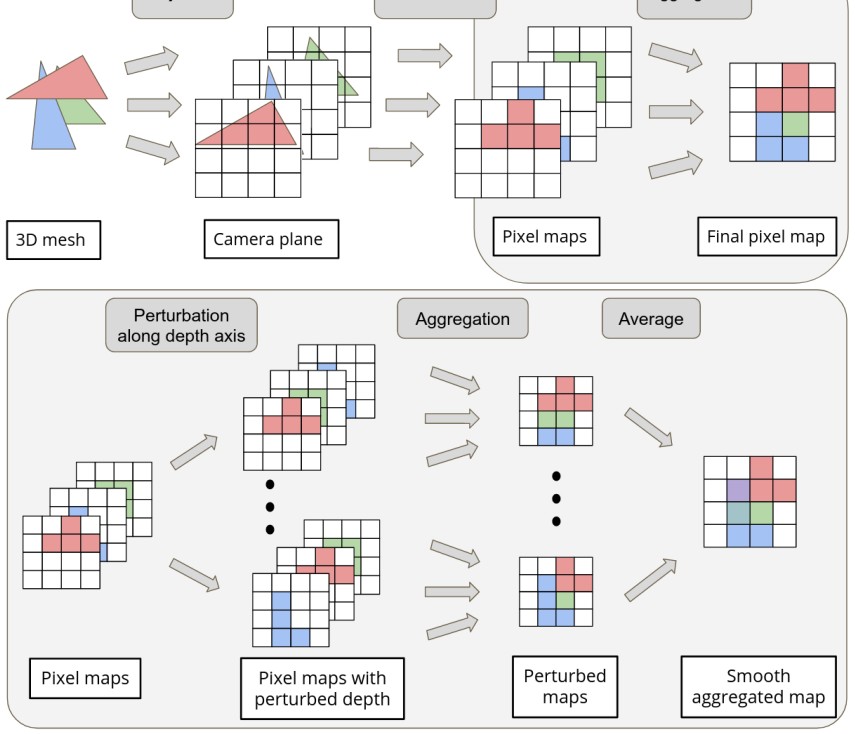

Figure 1: **Top**: Overview of the rendering process: both rasterization and aggregation steps induce non-smoothness in the computational flow. **Bottom:** Illustration of the differentiable perturbed aggregation process. The rasterization step is made differentiable in a similar way.

camera, some pixels will immediately change color at the moment the object enters the camera view or becomes unocluded by another object.

In this paper, we propose to exploit randomized smoothing techniques within the context of differentiable rendering to automatically soften the non-smooth rendering operations, making them naturally differentiable. The generality of our approach offers a theoretical understanding of some of the existing differentiable renderers while its flexibility leads to competitive or even state-of-the-art results in practice. We make the following contributions:

↪ We formulate the non-smooth operations occurring in the rendering process as solutions of optimization problems and, based on recent work [4], we propose a natural way of smoothing them using random perturbations. We highlight the versatility of this smoothing formulation and show how it offers a theoretical understanding of several existing differentiable renderers.

↪ We propose a general way to use control variate methods to reduce variance when estimating the gradients of the perturbed optimizers, which allows for sharp gradients even in the case of weaker perturbations.

↪ We introduce an adaptive scheme to automatically adjust the smoothing parameters by relying on sensitivity analysis of differentiable perturbed renderers, leading to a robust and adaptive behavior during the optimization process.

↪ We demonstrate through experiments on pose optimization and 3D mesh reconstruction that the resulting gradients combined to the adaptive smoothing provide a strong signal, leading to more accurate solutions when compared to state-of-the-art alternatives based on smooth gradient approximations [23].

## 2   Background

In this section, we review the fundamental aspects behind image rendering and recall the notion of perturbed optimizers which are at the core of the proposed approach.

**Rasterizer and "aggregater" as optimizers.** We consider the rendering process of an RGB image of height $h_I$ and width $w_I$, from a scene composed with meshes represented by a set of $m$ triangles. We assume that every triangle has already been projected onto the camera plane and their associated color maps $C_j \in \mathbb{R}^{h_I \times w_I \times 3}$, $j \in [1 \mathbin{..} m]$, have been computed using a chosen illumination model (Phong [29], Gouraud [13] etc.) and interpolating local properties of the mesh. We denote by $I$ the occupancy map such that $I_j^i$ is equal to 1 if the center of the $i^{th}$ pixel is inside the 2D projection of the $j^{th}$ triangle on the camera plane and 0 otherwise. By denoting $d(i,j)$ the Euclidean distance from the center of the $i^{th}$ pixel to the projection of the $j^{th}$ triangle, the rasterization step (which actually consists in computing the occupancy map $I$) can be written as:

$$I_j^i = H(d(i,j)), \tag{1}$$

where $H$ corresponds to the Heaviside function defined by:

$$H(x) = \left\{ \begin{array}{l} 0 \text{ if } x \leq 0 \\ 1 \text{ otherwise} \end{array} \right. = \operatorname*{argmax}_{0 \leq y \leq 1} y\, x. \tag{2}$$

We call $R \in \mathbb{R}^{h_I \times w_I \times 3}$ the final rendered image which is obtained by aggregating the color map of each triangle. In classical renderers, this step is done by using a Z-buffer so that only foreground objects are visible. This corresponds to:

$$R^i = \sum_j w_j^i(z) C_j^i, \;\; \text{with} \;\; w^i(z) = \operatorname*{argmax}_{y \text{ s.t. } \|y\|_1 = 1, y \geq 0, \; y_j = 0 \text{ if } I_j^i = 0} \langle z, y \rangle, \tag{3}$$

where $w, z \in \mathbb{R}^{m+1}$ and for $1 \leq j \leq m$, $z_j$ is the inverse depth of the $j^{th}$ triangle and the $(m+1)^{th}$ coordinate of $z$ and $w$ account for the background. The inverse depth of the background is fixed to $z_{m+1} = z_{\min}$. From Eq. (2) and (3), it appears that argmax operations play a central role in renderers and are typically non-differentiable functions with respect to their input arguments, as discussed next.

**Perturbed optimizers.** In [4], Berthet et al. introduce a generic approach to handle non-smooth problems of the form $y^*(\theta) = \operatorname*{argmax}_{y \in \mathcal{C}} \langle \theta, y \rangle$ with $\mathcal{C}$ a convex polytope, and make these problems differentiable by randomly perturbing them. More precisely, $y^*(\theta)$ necessarily lies on a vertex of the convex set $\mathcal{C}$. Thus, when $\theta$ is only slightly modified, $y^*$ remains on the same vertex, but when the perturbation grows up to a certain level, $y^*$ jumps onto another vertex. Concretely, this means that $y^*(\theta)$ is piece-wise constant with respect to $\theta$: the Jacobian $J_\theta y^*$ is null almost everywhere and undefined otherwise. This is why the rasterization (2) and aggregation (3) steps make renderers non-differentiable. Following [4], $y^*(\theta)$ can be approximated by the perturbed optimizer:

$$y_\epsilon^*(\theta) = \mathbb{E}_Z[y^*(\theta + \epsilon Z)] \tag{4}$$

$$= \mathbb{E}_Z \left[ \operatorname*{argmax}_{y \in \mathcal{C}} \langle \theta + \epsilon Z, y \rangle \right], \tag{5}$$

where $Z$ is a random noise following a distribution of the form $\mu(z) \propto \exp(-\nu(z))$. Then, we have $y_\epsilon^*(\theta) \xrightarrow{\epsilon \to 0} y^*(\theta)$, $y_\epsilon^*(\theta)$ is differentiable and its gradients are non-null everywhere. Intuitively, one can see from (5) that perturbed optimizers $y_\epsilon^*$ are actually a convolved version of the rigid optimizer $y^*$. Moreover, their Jacobian with respect to $\theta$ is given by:

$$J_\theta y_\epsilon^*(\theta) = \mathbb{E}_Z \left[ \frac{y^*(\theta + \epsilon Z)}{\epsilon} \nabla \nu(Z)^\top \right]. \tag{6}$$

It is worth noticing at this stage that, when $\epsilon = 0$, one recovers the standard formulation of the rigid optimizer. In general, $y_\epsilon^*(\theta)$ and $J_\theta y_\epsilon^*(\theta)$ do not admit closed-form expressions. To overcome this issue, Monte-Carlo estimators are exploited to compute an estimate of these quantities, as recalled in Sec. 4.

## 3 Related work

Our work builds on results in differentiable rendering, differentiable optimization and randomized smoothing.

**Differentiable rendering.** Some of the earliest differentiable renderers rely on the rigid rasterization-based rendering process recalled in the previous section, while exploiting some gradient approximations of the non-smooth operations in the backward. OpenDR [24] uses a first-order Taylor expansion to estimate the gradients of a classical renderer, resulting in gradients concentrated around the edges of the rendered images. In the same vein, NMR [16] proposes to avoid the issue of local gradients by doing manual interpolation during the backward pass. For both OpenDR and NMR, the discrepancy between the function evaluated in the forward pass and the gradients computed in the backward pass may lead to unknown or undesired behaviour when optimizing over these approximated gradients. More closely related to our work, SoftRas [23] directly approximates the forward pass of usual renderers to make it naturally differentiable. Similarly, DIB-R [8] introduces the analytical derivatives of the faces' barycentric coordinates to get a smooth rendering of foreground pixels during the forward pass, and thus, gets gradients naturally relying on local properties of meshes. Additionally, a soft aggregation step is required to backpropagate gradients towards background pixels in a similar way to SoftRas. In a parallel line of work, physically realistic renderers are made differentiable by introducing a stochastic estimation of the derivatives of the ray tracing integral [21, 28]. This research direction seems promising as it allows to generate images with global illumination effects. However, this class of renderers remains computationally expensive when compared to rasterization-based algorithms, which makes them currently difficult to exploit within classic computer vision or robotics contexts, where low computation timings matter.

**Differentiable optimization and randomized smoothing.** More generally, providing ways to differentiate solutions of constrained optimization problems has been part of the recent growing effort to introduce more structured operations in the differentiable programming paradigm [33], going beyond classic neural networks. A first approach consists in automatically unrolling the optimization process used to solve the problem [11, 26]. However, because the computational graph grows with the number of optimization steps, implicit differentiation, which relies on the differentiation of optimality conditions, should be preferred when available [2, 3, 20]. These methods compute *exact* gradients whenever the solution is differentiable with respect to the parameters of the problem. In this paper, we show how differentiating through a classic renderer relates to computing derivatives of a Linear Programming (LP) problem. In this particular case and as recalled in the Sec. 2, solutions of the optimization problem vary in a heavily non-smooth way as gradients are null almost-everywhere and non-definite otherwise, thus, making them hard to exploit within classic optimization algorithms. To overcome this issue, a first approach consists in introducing a gradient proxy by approximating the solution with a piecewise-linear interpolation [35] which can also lead to null gradient. Another approach leverages random perturbations to replace the original problem by a smoother approximation [1, 4]. For our differentiable renderer, we use this later method as it requires only little effort to transform non-smooth optimizers into differentiable ones, while guaranteeing non-null gradients everywhere. Moreover, randomized smoothing has been shown to facilitate optimization of non-smooth problems [12]. Solving a smooth approximation leads to improved performance when compared to methods dealing with the original rigid problem. In addition to its smoothing effect, the addition of noise also acts as an implicit regularization during the training process which is valuable for globalization aspects [4, 9], i.e. converging towards better local optima.

## 4 Approximate differentiable rendering via perturbed optimizers

In this section, we detail the main contributions of the paper. We first introduce the reformulation of the rasterization and aggregation steps as perturbed optimizers, and propose a variance reduction mechanism to alleviate the computational burden inherent to these kinds of stochastic estimators. Additionally, we introduce an adaptive scheme to automatically adjust the smoothing parameters inherent to the rendering process.

### 4.1 Perturbed renderer: a general approximate and differentiable renderer

As detailed in Sec. 2, the rasterization step (2) can be directly written as the argmax of an LP. Similarly, the aggregation (3) step can be slightly modified to reformulate it as an argmax of an LP. Indeed, when $y \geq 0$, the hard constraint $y_j = 0$ if $I_j^i = 0$ is equivalent to $I_j^{i y_j} > 0$, which can be

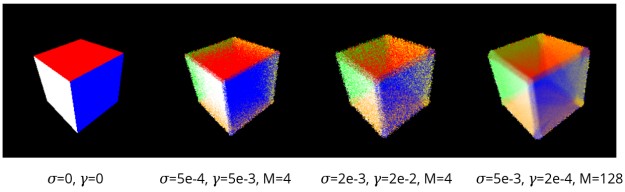

$\sigma=0, \gamma=0$      $\sigma=5e\text{-}4, \gamma=5e\text{-}3, M=4$      $\sigma=2e\text{-}3, \gamma=2e\text{-}2, M=4$      $\sigma=5e\text{-}3, \gamma=2e\text{-}4, M=128$

Figure 2: Examples of images obtained from a perturbed differentiable rendering process with a Gaussian noise. With smoothing parameters set to 0, we retrieve the rigid renderer, while adding noise makes pixels from the background appear on the foreground and vice versa.

approximated by adding a logarithmic barrier $y_j \ln I_i^j$ in the objective function as done in classical interior point methods [6]:

$$w_\alpha^i(z) = \operatorname*{argmax}_{y \ s.t. \ \|y\|_1=1, y\geq 0} \langle z + \frac{1}{\alpha}\ln(I^i), y \rangle, \tag{7}$$

because $\ln I_j^i \xrightarrow{I_j^i \to 0} -\infty$, enforcing $w_j^i \xrightarrow{I_j^i \to 0} 0$. $\alpha \to +\infty$ approximates the hard constraint and allows to retrieve the classical formulation (3) of $w$. Using this formulation, it is possible to introduce a differentiable approximation of the rasterization and aggregation steps:

$$\hat{I}_j^i = H_\sigma(d(i,j)), \text{ with } H_\sigma(x) = \mathbb{E}_X[H(x+\sigma X)] = \mathbb{E}_X[\operatorname*{argmax}_{0\leq y\leq 1} \langle y, x+\sigma X \rangle], \tag{8}$$

$$\hat{R}^i = \sum_j w_{\alpha, \gamma_j}^i(z)C_j^i, \text{ with } w_{\alpha,\gamma}^i(z) = \mathbb{E}_Z[w_\alpha(z+\gamma Z)] \tag{9}$$

$$= \mathbb{E}_Z[\operatorname*{argmax}_{y \ s.t. \ \|y\|_1=1, y\geq 0} \langle y + \frac{1}{\alpha}\ln(\hat{I}^i), z+\gamma Z \rangle]. \tag{10}$$

By proceeding this way, we get a rendering process for which every pixel is influenced by every triangle of meshes (Fig. 1,2), similarly to SoftRas [23]. More concretely, as shown in [4], the gradients obtained with the randomized smoothing are guaranteed to be non-null everywhere unlike some existing methods [8, 16, 24]. This makes it possible to use the resulting gradients as a strong supervision signal for optimizing directly at the pixel level, as shown through the experiments in Sec. 5. At this stage, it is worth noting that the prior distribution on the noise $Z$ used for smoothing is not fixed. One can choose various distributions as a prior thus leading to different smoothing patterns, which makes the approach versatile. Indeed, as a different noise prior induces different smoothing, some specific choices allow to retrieve some of the existing differentiable renderers [16, 23, 24]. In particular, using a Logistic and a Gumbel prior respectively on $X$ and $Z$ leads to SoftRas[23]. Further details are included in Appendix A.

More practically, the smoothing parameters $\sigma, \gamma$ appearing in differentiable renderers such as SoftRas [23] or DIB-R [8] can be hard to set. In contrast, in the proposed approach, the smoothing is naturally interpretable as a noise acting directly on positions of mesh triangles. Thus, the smoothing parameters representing the noise intensity can be scaled automatically according to the object dimensions. In Sec. 4.3, we notably propose a generic approach to automatically adapt them along the optimization process.

### 4.2 Exploiting the noise inside a perturbed renderer

In practice, (5) and (6) are useful even in the cases when the choice of prior on $Z$ does not induce any analytical expression for $y_\epsilon^*$ and $J_\theta y_\epsilon^*$. Indeed, Monte-Carlo estimators of these two quantities can be directly obtained from these two expressions:

$$y_\epsilon^M(x) = \frac{1}{M}\sum_{i=1}^M y^*(\theta + \epsilon Z^{(i)}) \tag{11}$$

$$J_\theta y_\epsilon^M(z) = \frac{1}{M}\sum_{i=1}^M y^*(\theta + \epsilon Z^{(i)})\frac{1}{\epsilon}\nabla\nu(Z^{(i)})^\top, \tag{12}$$

Table 1: Time and memory complexity of our perturbed renderer for the forward and backward computation during the pose optimization task (5).

| # of samples | 1 | 2 | 8 | 32 | 64 | SoftRas | Hard renderer |
|---|---|---|---|---|---|---|---|
| Forward (ms) | 30 ($\pm 3$) | 30 ($\pm 3$) | 31 ($\pm 3$) | 31 ($\pm 3$) | 31 ($\pm 3$) | 29 ($\pm 3$) | 29 ($\pm 3$) |
| Backward (ms) | 19 ($\pm 1$) | 19 ($\pm 1$) | 19 ($\pm 2$) | 20 ($\pm 1$) | 22 ($\pm 1$) | 18 ($\pm 1$) | N/A |
| Max memory (Mb) | 217.6 | 217.6 | 217.6 | 303.1 | 440.0 | 180.3 | 178.5 |

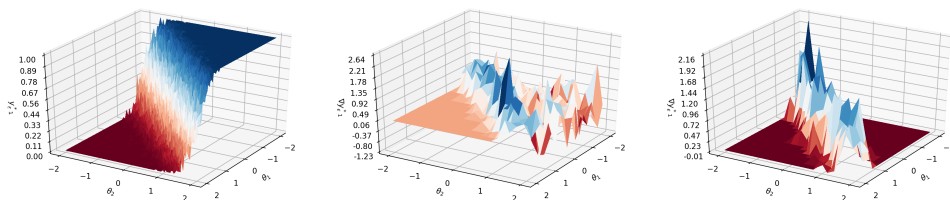

Figure 3: Control variates methods [19] allow to reduce the variance of the gradient of the smooth argmax. **Left**: Gaussian-perturbed argmax operator, **Middle**: estimator of gradient **Right**: variance-reduced estimator of gradient.

where $M$ is the total number of samples used to compute the approximations.

Approximately, we have $\text{Var}\left[J_\theta y_\epsilon^M(z)\right] \propto \frac{\text{Var}\left[y^*(\theta + \epsilon Z)\nabla\nu(Z^{(i)})^\top\right]}{M\epsilon^2}$. When decreasing $\epsilon$, the variance of the Monte-Carlo estimator increases, which can make the gradients very noisy and difficult to exploit in practice. For this reason, we use the control variates method [19] in order to reduce the variance of our estimators. Because the quantity $y^*(\theta)\nabla\nu(Z)^\top$ has a null expectation when $Z$ has a symmetric distribution and is positively correlated with $y^*(\theta + \epsilon Z)\nabla\nu(Z)^\top$, we can rewrite (6) as:

$$J_\theta y_\epsilon^*(\theta) = \mathbb{E}_Z\left[(y^*(\theta + \epsilon Z) - y^*(\theta))\nabla\nu(Z)^\top/\epsilon\right],\qquad(13)$$

which naturally leads to a variance-reduced Monte-Carlo estimator of the Jacobian:

$$J_\theta y_\epsilon^M(\theta) = \frac{1}{M}\sum_{i=1}^M\left(y^*(\theta + \epsilon Z^{(i)}) - y^*(\theta)\right)\frac{1}{\epsilon}\nabla\nu(Z^{(i)})^\top.\qquad(14)$$

The computation of $y_\epsilon^M$ requires the solution of $M$ perturbed problems instead of only one in the case of classical rigid optimizer. Fortunately, this computation is naturally parallelizable, leading to a constant computation time at the cost of an increased memory footprint. Using our variance-reduced estimator alleviates the need for a high number of Monte Carlo samples, hence reducing the computational burden inherent to perturbed optimizers (Fig. 3). As shown in Sec. 5 (Fig. 5, left), $M = 8$ samples is already sufficient to get stable and accurate results. In addition, computations from the forward pass can be reused in the backward pass and time and memory complexities to evaluate these passes are comparable to classical differentiable renderers (Tab. 1). Consequently, evaluating gradients does not require any extra computation. It is worth mentioning at this stage that this variance reduction mechanism in fact applies to any perturbed optimizer. This variance reduction technique can be interpreted as estimating a sub-gradient with the finite differences in a random direction and is inspired by the field of random optimization [27].

### 4.3 Making the smoothing adaptive with sensitivity analysis

In a way similar to (6), it is possible to formulate the sensitivity of the perturbed optimizer with respect to the smoothing parameter $\epsilon$:

$$J_\epsilon y_\epsilon^*(\theta) = \mathbb{E}_Z\left[y^*(\theta + \epsilon Z)\left(\nabla\nu(Z)^\top Z - 1\right)/\epsilon\right]\qquad(15)$$

and, as previously done, we can build a variance-reduced Monte-Carlo estimator of this quantity:

$$J_\epsilon y_\epsilon^M(\theta) = \frac{1}{M}\sum_{i=1}^M\left(y^*(\theta + \epsilon Z^{(i)}) - y^*(\theta)\right)\frac{\nabla\nu(Z^{(i)})^\top Z^{(i)} - 1}{\epsilon}\qquad(16)$$

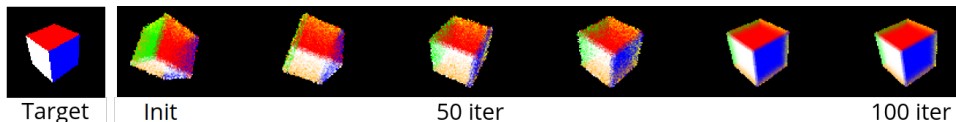

Target    Init    50 iter    100 iter

Figure 4: Perturbed differentiable renderer and adaptive smoothing lead to precise 6D pose estimation.

In the case of differentiable rendering, a notable property is the positivity of the sensitivity of any locally convex loss $\mathcal{L}$ (which is valid for the RGB loss) with respect to the $\gamma$ smoothing parameter when approaching the solution. Indeed, in the neighborhood of the solution, we have $\theta \approx \theta_t$ and a first-order Taylor expansion gives:

$$\frac{\partial \mathcal{L}}{\partial \gamma}(\theta) \approx J_\gamma w_\gamma(z)^\top C^\top \nabla^2 \mathcal{L}(R(\theta_t)) C J_\gamma w_\gamma(z) \geq 0, \tag{17}$$

because for a locally convex loss, $\nabla^2 \mathcal{L}(R(\theta_t))$ is positive definite near a local optimum. We exploit this property to gradually reduce the smoothing when needed. To do so, we track an exponential moving average of the sensitivity during the optimization process:

$$v_\gamma^t = \beta_\gamma v_\gamma^{t-1} + (1 - \beta_\gamma) \frac{\partial \mathcal{L}}{\partial \gamma}(\theta_t) \tag{18}$$

where $\beta_\gamma$ is a scalar parameter. Whenever $v_\gamma^t$ is positive, the smoothing $(\sigma, \gamma)$ is decreased at a constant rate (Fig. 4). Note that the adaptive algorithm does not require any further computation as sensitivity can be obtained from backpropagation and also beneficially applies to other differentiable renderers [23] (see 5).

## 5   Results

In this section, we explore applications of the proposed differentiable rendering pipeline to standard computer vision tasks: single-view 3D pose estimation and 3D mesh reconstruction. Our implementation is based on Pytorch3d [31] and will be publicly released upon publication. It corresponds to a modular differentiable renderer where switching between different type of noise distributions (Gaussian, Cauchy, etc.) is possible. We notably compare our renderer against the open source SoftRas implementation available in Pytorch3d and DIB-R from Kaolin library. The results for these renderers are obtained by running the experiments in our setup as we were not able to get access to the original implementations of the pose optimization and shape reconstruction problems.

**Single-view 3D pose estimation** is concerned with the retrieving the 3D rotation $\theta \in SO(3)$ of an object from a reference image. Similarly to [23], we first consider the case of a colored cube and fit the rendered image of this cube to a single view of the true pose $R(\theta_t)$ (Fig. 4). The problem can be formulated as a regression problem of the form:

$$\min_\theta \mathcal{L}_{RGB}(\theta) = \frac{1}{2} \|R(\theta_t) - \hat{R}(\theta)\|_2^2, \tag{19}$$

where $\hat{R}$ is the smooth differentiable rendered image. We aim at analysing the sensitivity of our perturbed differentiable renderer with respect to random initial guesses of various amplitudes from the real pose. Thus, the initial $\theta$ is randomly perturbed from the true pose with various intensity for the perturbation (exploiting an angle-axis representation of the rotation, with a randomized rotation axis). The intensity of the angular perturbation is taken between 20 and 80 degrees. Consequently, the value of the average final error has an increased variance and is often perturbed by local minima (Tab. 2). To avoid these caveats, we rely on another metric: the percentage of solved tasks (see Tab. 2 and Fig. 5 and 6), accounting for the amount of final errors which are below a given threshold (10 degrees for Tab. 2 and bottom row of Fig. 6). For this task, we compare our method with two different distributions for the smoothing noise (Gaussian and Cauchy priors) to SoftRas. We use Adam [17] with parameters $\beta_1 = 0.9$ and $\beta_2 = 0.999$ and operate with $128 \times 128$ RGB images, each optimization problem taking about 1 minute to solve on a Nvidia RTX6000 GPU.

Table 2: Results of pose optimization from single-view image using perturbed differentiable renderer with variance reduction and adaptive smoothing.

| Initial error | 20° | | | 50° | | | 80° | | |
|---|---|---|---|---|---|---|---|---|---|
| Diff. renderer | SoftRas | Cauchy smoothing | Gaussian smoothing | SoftRas | Cauchy smoothing | Gaussian smoothing | SoftRas | Cauchy smoothing | Gaussian smoothing |
| Avg. error (°) | 5.5 | 6.1 | **3.8** | 14.6 | 20.9 | **11.6** | 33.5 | 41.8 | **25.3** |
| Std. error (°) | 4.9 | 5.1 | 4.4 | 26.6 | 21.0 | 22.5 | 35.3 | 34.0 | 34.9 |
| Task solved (%) | 84 ±4.0 | 82 ±2.3 | **93** ±0.4 | 75 ±5.3 | 71 ±2.5 | **84** ±1.3 | 55 ±4.8 | 55 ±5.5 | **67** ±2.6 |

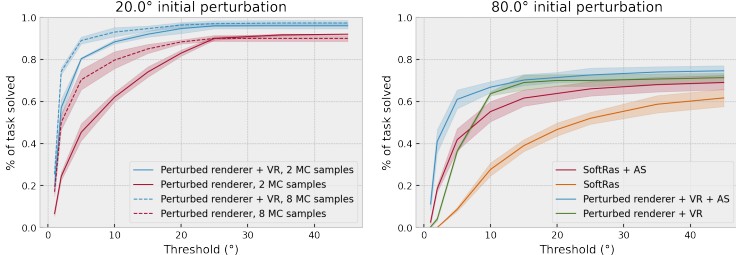

Figure 5: **Left**: Variance reduction (VR) drastically reduces the number of Monte-Carlo samples required to retrieve the true pose from a 20° perturbation. **Right**: Adaptive smoothing (AS) improves resolution of precise pose optimization for a 80° perturbation.

We provide results of 100 random perturbations for various magnitude of initial perturbations in Tab. 2 and Fig. 5,6. Error bars provide standard deviations while running experiments with different random seeds. Fig. 5 (left) demonstrates the advantage of our variance reduction method to estimate perturbed optimizers and precisely retrieve the true pose at a lower computational cost. Indeed, the variance-reduced perturbed renderer is able to perform precise optimization with only 2 samples (80% of final errors are under 5° while it is less than 50% without the use of control variates). Additionally, our adaptive smoothing significantly improves the number of solved tasks for both SoftRas and our perturbed renderer (Fig. 5, right).

Additional results in Tab. 2, Fig. 6 and Fig. 11 in Appendix demonstrate that our perturbed renderers with a Gaussian smoothing, outperforms SoftRas and can achieve state-of-the-art results for pose optimization tasks on various objects. As shown in Appendix (Prop. 4), SoftRas is equivalent to using a Gumbel smoothing which is an asymmetric prior. However, no direction should be preferred when smoothing the rendering process which explain the better performances of the Gaussian prior.

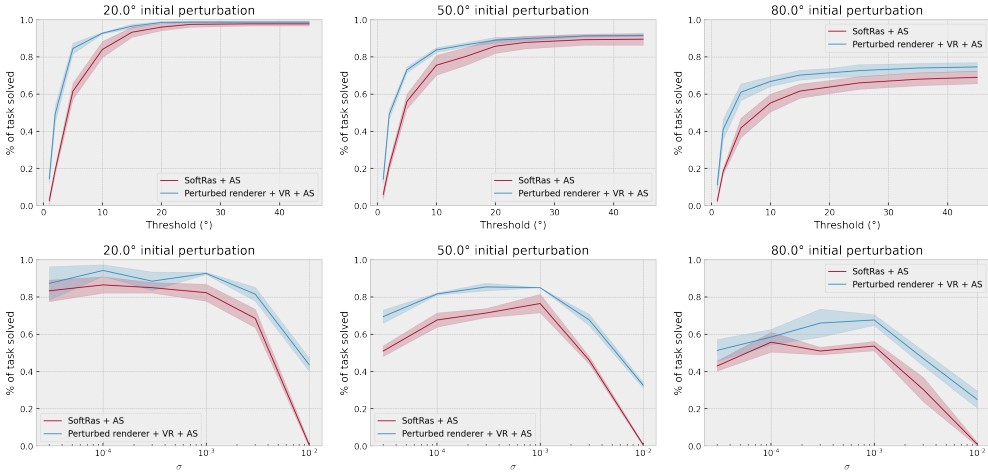

Figure 6: **Top row**: perturbed renderer combined with variance reduction and adaptive scheme improves SoftRas results on pose optimization. **Bottom row**: The method is robust w.r.t. initial smoothing values. Higher is better.

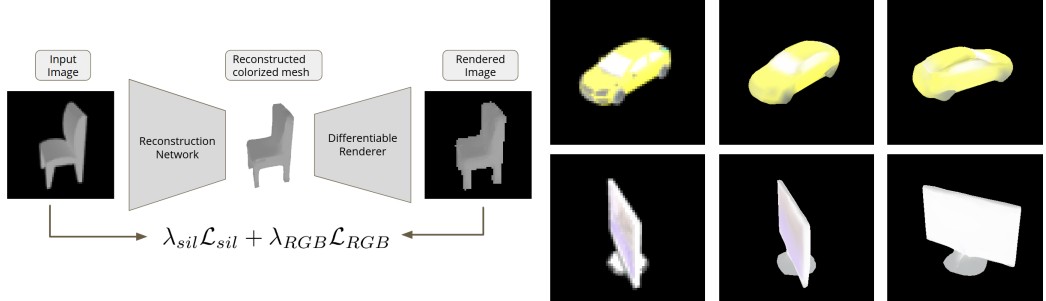

Figure 7: **Left**: A neural network is trained with a self-supervision signal from a differentiable rendering process. **Right**: Qualitative results from self-supervised 3D mesh reconstruction using a perturbed differentiable renderer. **1ˢᵗ column**: Input image. **2ⁿᵈ and 3ʳᵈ columns**: reconstructed mesh viewed from different angles.

Table 3: Results of mesh reconstruction on the ShapeNet dataset [7] reported with 3D IoU (%)

| | Airplane | Bench | Dresser | Car | Chair | Display | Lamp | Speaker | Rifle | Sofa | Table | Phone | Vessel ‖ | Mean |
|---|---|---|---|---|---|---|---|---|---|---|---|---|---|---|
| NMR [16] | 58.5 | 45.7 | 74.1 | 71.3 | 41.4 | 55.5 | 36.7 | 67.4 | 55.7 | 60.2 | 39.1 | 76.2 | 59.4 | 57.0 |
| SoftRas [23] [1] | 62.0 | 47.55 | 66.2 | 69.4 | 49.4 | 60.0 | 43.3 | 62.6 | 61.4 | 60.4 | 43.6 | 76.4 | 59.9 | 58.6 |
| DIB-R [8] [2] | 59.7 | **50.9** | 66.2 | **72.6** | **52.0** | 57.9 | 43.8 | **63.8** | 61.0 | 65.4 | **50.5** | 76.3 | 58.6 | 59.9 |
| Ours | **63.5** | 49.4 | **67.1** | 72.3 | 51.6 | **60.4** | **44.3** | 62.8 | **65.7** | **66.7** | 49.2 | **80.2** | **60.0** | 61.0 |
| | (±0.15) | (±0.08) | (±0.32) | (±0.15) | (±0.11) | (±0.14) | (±0.31) | (±0.24) | (±0.09) | (±0.12) | (±0.21) | (±0.33) | (±0.07) | (±0.11) |

Bottom row of Fig. 6 shows that our method is robust to the choice of initial values of smoothing parameters. During optimization, the stochasticity of the gradient due to the noise from the perturbed renderer acts as an implicit regularization of the objective function. This helps to avoid sharp local minima or saddle points and converge to more stable minimal regions, leading to better optima. As mentioned earlier, results are limited by the inherent non-convexity of the rendering process and differentiable renderers are not able to recover from a bad initial guess. Increasing the noise intensity to further smoothen the objective function would not help in this case, as it would also lead to a critical loss of information on the rendering process.

**Self-supervised 3D mesh reconstruction from a single image.** In our second experiment, we demonstrate the ability of our perturbed differentiable renderer to provide a supervisory signal for training a neural network. We use the renderer to self-supervise the reconstruction of a 3D mesh and the corresponding colors from a single image (see Fig. 7). To do so, we use the network architecture proposed in [23] and the subset of the Shapenet dataset [7] from [16]. The network is composed of a convolutional encoder followed by two decoders: one for the mesh reconstruction and another one for color retrieving (more details in Appendix B).

Using the same setup as described in [8, 16, 23] , the training is done by minimizing the following loss:

$$\mathcal{L} = \lambda_{sil}\mathcal{L}_{sil} + \lambda_{RGB}\mathcal{L}_{RGB} + \lambda_{lap}\mathcal{L}_{lap},$$ (20)

where $\mathcal{L}_{sil}$ is the negative IoU between silhouettes $I$ and $\hat{I}$, $\mathcal{L}_{RGB}$ is the $\ell_1$ RGB loss between $R$ and $\hat{R}$, $\mathcal{L}_{lap}$ is a Laplacian regularization term penalizing the relative change in position between neighbour vertices (detailed description in Appendix B). For training, we use Adam algorithm with a learning rate of $10^{-4}$ and parameters $\beta_1 = 0.9, \beta_2 = 0.999$. Even if the memory consumption of the perturbed renderer may limit the maximum size of batches, its flexibility makes it possible to switch to a deterministic approximation such as [23] to use larger batches in order to finetune the neural network, if needed.

We analyze the quality of obtained 3D reconstruction through a rendering pipeline by providing IoU measured on 13 different classes of the test set (see Tab. 3). We observe that our perturbed

---

[1]These numbers were obtained by running the renderer in our own setup (cameras, lightning, training parameters etc.) in order to have comparable results. This may explain the slight difference with the numbers from the original publications.

renderer obtains state-of-the-art accuracies on this task. Error bars (representing the standard deviation obtained by retraining the model with different random seeds) confirm the stability of our method. Additionally, we illustrate qualitative results for colorized 3D mesh reconstruction from a single image in Fig. 7(right) confirming the ability of perturbed renderers to provide strong visual supervision signals.

## 6 Conclusion

In this paper, we introduced a novel approach to differentiable rendering leveraging recent contributions on perturbed optimizers. We demonstrated the flexibility of our approach grounded in its underlying theoretical formulation. The combination of our proposed perturbed renderers with variance-reduction mechanisms and sensitivity analysis enables for robust application of our rendering approach in practice. Our results notably show that perturbed differentiable rendering can reach state-of-the-art performance on pose optimization and is also competitive for self-supervised 3D mesh reconstruction. Additionally, perturbed differentiable rendering can be easily implemented on top of existing implementations without major modifications, while generalizing existing renderers such as SoftRas. As future work, we plan to embed our generic differentiable renderer within a differentiable simulation pipeline in order to learn physical models, by using visual data as strong supervision signals to learn physics.

## Acknowledgments and Disclosure of Funding

We warmly thank Francis Bach for useful discussions. This work was supported in part by the HPC resources from GENCI-IDRIS(Grant AD011012215), the French government under management of Agence Nationale de la Recherche as part of the "Investissements d'avenir" program, reference ANR-19-P3IA-0001 (PRAIRIE 3IA Institute), and Louis Vuitton ENS Chair on Artificial Intelligence.

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
