# Appendices

## A  Differentiable perturbed renderer

This section describes some possible choice of priors for the smoothing noise of our differentiable perturbed renderers and discuss the eventual links with already existing renderers. In addition, we provide justification of the variance reduction and adaptive smoothing methods for different priors on the noise.

### A.1  Perturbed renderers

The following propositions discuss some possible choices of noise prior for the perturbed renderer, exhibiting the links with already existing differentiable renderers [24, 16, 23].

**Proposition 1.** *The sigmoid function corresponds to the perturbed Heaviside function with a logistic prior on noise:*

$$sigmoid(x) = \mathrm{E}[H(x + Z)] \tag{21}$$

*where $Z \sim Logistic(0, 1)$*

*Proof.* First, we note that the sigmoid function correspond to the first weight from the softmax applied to the $\mathrm{R}^2$ vector $\widetilde{x} = \begin{pmatrix} x \\ 0 \end{pmatrix}$. Indeed, we have :

$$\sigma(x) = \frac{1}{1 + \exp(-x)} = \frac{\exp(x)}{\exp(x) + \exp(0)} \tag{22}$$

$$= softmax\,(\widetilde{x})^T \begin{pmatrix} 1 \\ 0 \end{pmatrix} \tag{23}$$

From the previous proposition, with $Z$ a random variable in $\mathrm{R}^2$ distributed with a Gumbel distribution, we have:

$$softmax(\widetilde{x}) = \mathrm{E}[\operatorname*{argmax}_{y \ s.t. \ \|y\|_1 = 1} \langle y, \widetilde{x} + \epsilon Z \rangle] \tag{24}$$

$$= \mathrm{E}[\operatorname*{argmax}_{y \ s.t. \ \|y\|_1 = 1} y^T \begin{pmatrix} x + \epsilon Z_1 \\ \epsilon Z_2 \end{pmatrix}] \tag{25}$$

$$= \mathrm{E}[\operatorname*{argmax}_{y \ s.t. \ \|y\|_1 = 1} y^T \begin{pmatrix} x + \epsilon(Z_1 - Z_2) \\ 0 \end{pmatrix}] \tag{26}$$

because $x + \epsilon(Z_1 - Z_2) > 0$ is equivalent to $x + \epsilon Z_1 > \epsilon Z_2$.

Finally, we can re-write:

$$\sigma(x) = \mathrm{E}[\operatorname*{argmax}_{y \ s.t. \ \|y\|_1 = 1} y^T \begin{pmatrix} x + \epsilon \widetilde{Z} \\ 0 \end{pmatrix}]^T \begin{pmatrix} 1 \\ 0 \end{pmatrix} \tag{27}$$

$$= \mathrm{E}[\operatorname*{argmax}_{0 \le y \le 1} y(x + \epsilon \widetilde{Z})] \tag{28}$$

$$= \mathrm{E}[H(x + \epsilon \widetilde{Z})] \tag{29}$$

where $\widetilde{Z}$ follows a logistic law.

Which allows to conclude that the sigmoid corresponds to the Heaviside function perturbed with a logistic noise. We could use the fact that the cumulative distribution function of the logistic distribution is the sigmoid function to make an alternative proof. $\qquad\square$

**Proposition 2.** *The affine approximation of the step function corresponds to the perturbed Heaviside function with an uniform prior on noise:*

$$H_{aff}(x) = \mathrm{E}[H(x + Z)] \tag{30}$$

*where $Z \sim \mathcal{U}(-\frac{1}{2}, \frac{1}{2})$*

*Proof.* We have :

$$\mathrm{E}\left[H(x + Z)\right] = \mathrm{P}(x + Z > 0) \tag{31}$$

$$= \mathrm{P}(Z > -x) \tag{32}$$

$$= \begin{cases} 0 & if\ x \leq -\frac{1}{2} \\ x + \frac{1}{2} & if\ -\frac{1}{2} < x < \frac{1}{2} \\ 1 & if\ x \geq \frac{1}{2} \end{cases} \tag{33}$$

$$= H_{aff}(x) \tag{34}$$

$\square$

**Proposition 3.** *The arctan approximation of the step function corresponds to the perturbed Heaviside function with a Cauchy prior on noise:*

$$\frac{1}{2} + \frac{1}{\pi}arctan(x) = \mathrm{E}[H(x + Z)] \tag{35}$$

*where* $Z \sim Cauchy(0,1)$

*Proof.* We have :

$$\mathrm{E}\left[H(x + Z)\right] = \mathrm{P}(x + Z > 0) \tag{36}$$

$$= \mathrm{P}(Z > -x) \tag{37}$$

$$= \int_{-x}^{\infty} \frac{1}{\pi(1 + x^2)} dz \tag{38}$$

$$= \frac{1}{2} + \frac{1}{\pi}arctan(x) \tag{39}$$

$\square$

**Proposition 4.** *The softmax function corresponds to the perturbed maximal coordinates with a Gumbel prior on the noise:*

$$softmax(z) = \mathrm{E}[\underset{y\ s.t.\ \|y\|_1 = 1}{argmax} \langle y, z + Z \rangle] \tag{40}$$

*where* $Z \sim Gumbel(0,1)$

*Proof.* The Gumbel-max trick is a classical property of the Gumbel distribution stating: the maximizer of M fixed values $z_i$ which have been perturbed by a noise i.i.d $Z_i$ following a Gumbel distribution, follows an exponentially weighted distribution [14]. This can be written:

$$\mathrm{P}(\max_j z_j + Z_j = z_i + Z_i) = \frac{\exp(z_i)}{\sum_j \exp(z_j)} \tag{41}$$

where $Z_i$ are i.i.d. and following a Gumbel distribution. Thus, for a noise $Z$ following a Gumbel distribution, we have :

$$\mathrm{E}[\underset{y\ s.t.\ \|y\|_1 = 1}{argmax} \langle y, z + \epsilon Z \rangle] = \sum_i e_i \mathrm{P}(\underset{y\ s.t.\ \|y\|_1 = 1}{argmax} \langle y, z + \epsilon Z \rangle = e_i) \tag{42}$$

$$= \sum_i e_i \mathrm{P}(\max_j z_j + Z_j = z_i + \epsilon Z_i) \tag{43}$$

$$= \sum_i e_i \frac{\exp(z_i)}{\sum_j \exp(z_j)} \tag{44}$$

$$= softmax(z) \tag{45}$$

$\square$

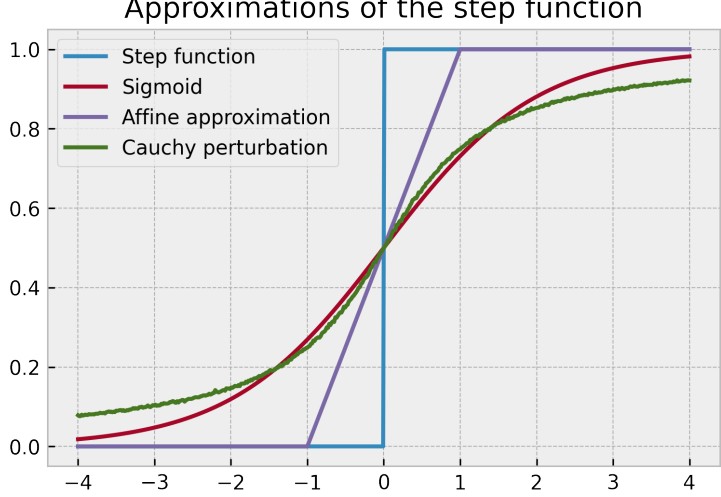

Figure 8: Modifying the prior on the noise leads to different approximations of a non-smooth operator. The sigmoid and affine approximations are obtained by using respectively a Logistic and a Uniform prior.

Following the previous results, one can observe that OpenDR [24] corresponds to a rasterization step where a uniform prior is used for the smoothing noise in order to get gradients during the backward pass. More precisely, the uniform distribution is taken with a support as large as a pixel size which allows to get non null gradients for pixels lying on the border of triangles. This can be linked to antialiasing techniques which are added to classical rendering pipeline to remove high frequencies introduced by the non-smooth rasterization. These techniques make the intensity of a pixel vary linearly with respect to the position of the triangle which can also be seen as an affine approximation of the rasterization. However, one should note that the density of the uniform distribution cannot be written in the form of $\mu(z) \propto \exp(-\nu(z))$ so the properties on differentiability from [4] does not apply. In particular, this approximation is non-differentiable at some points ($x = 1$ or $-1$) and can have null gradients (for $x < -1$ or $x > 1$) which is inducing localized gradients for OpenDR [24]. For this reason, the approach of NMR [16] actually consists in having a variable range for the distribution support in order to get non-null gradients for a wider region. This approach corresponds to the affine interpolation proposed in [35] and can lead to null gradients. More closely related to our approach, SoftRas smoothen the rasterization by using a sigmoid approximation while Z-buffering is replaced by a softmax during aggregation. As shown by propositions 1 and 4, this approach actually corresponds to a special case of a perturbed renderer where rasterization and aggregation steps were smoothen by using respectively a Logistic and a Gumbel prior for the noise.

In short, this means that some existing renderers [24, 23, 16] can be interpreted in the framework of differentiable perturbed renderers when considering specific priors for the noise used for smoothing. In addition, the generality of the approach allows to consider a continuous range of novel differentiable renderers in between the already existing ones.

### A.2   Variance reduction

As introduced in the paper, control variates methods can be used to reduce the noise of the Monte-Carlo estimators of the Jacobian of a perturbed renderer, without inducing any extra-computation. In particular, we prove this in the case of Gaussian and Cauchy priors on the smoothing noise.

**Proposition 5.** *Jacobian of perturbed renderers can be written as :*

$$J_\theta y_\epsilon^*(\theta) = \mathbb{E}_Z \left[ (y^*(\theta + \epsilon Z) - y^*(\theta)) \nabla \nu(Z)^\top \frac{1}{\epsilon} \right], \tag{46}$$

*when $Z$ follows a Gaussian distribution.*

*Proof.* As done in [4, 1], with a change of variable $z' = \theta + \epsilon z$ we have:

$$J_\theta y_\epsilon^*(\theta) = \frac{\partial}{\partial \theta} \left( \int_{-\infty}^{\infty} y^*(\theta + \epsilon z) \mu(z) dz \right), \tag{47}$$

$$= \int_{-\infty}^{\infty} y^*(z') \frac{\partial}{\partial \theta} \mu \left( \frac{z' - \theta}{\epsilon} \right) dz', \tag{48}$$

$$= \int_{-\infty}^{\infty} y^*(z') \frac{1}{\epsilon} \nabla \nu \left( \frac{z' - \theta}{\epsilon} \right)^\top \mu \left( \frac{z' - \theta}{\epsilon} \right) dz' \tag{49}$$

And by doing the inverse change of variable, we get:

$$J_\theta y_\epsilon^*(\theta) = \mathbb{E}_Z \left[ \frac{y^*(\theta + \epsilon Z)}{\epsilon} \nabla \nu(Z)^\top \right] \tag{50}$$

In addition, for $Z$ following a standard Gaussian distribution we have $\nu(Z) = \frac{\|Z\|^2}{2}$ and thus:

$$\mathbb{E}_Z \left[ y^*(\theta) \nabla \nu(Z)^\top \right] = y^*(\theta) \mathbb{E}_Z \left[ Z^\top \right] \tag{51}$$

and because $Z$ is centered, we get:

$$\mathbb{E}_Z \left[ y^*(\theta) \nabla \nu(Z)^\top \right] = 0 \tag{52}$$

Finally we get:

$$J_\theta y_\epsilon^*(\theta) = \mathbb{E}_Z \left[ \left( y^*(\theta + \epsilon Z) - y^*(\theta) \right) \nabla \nu(Z)^\top \frac{1}{\epsilon} \right]. \tag{53}$$

$\square$

**Proposition 6.** *Jacobian of perturbed renderers can be written as :*

$$J_\theta y_\epsilon^*(\theta) = \mathbb{E}_Z \left[ \left( y^*(\theta + \epsilon Z) - y^*(\theta) \right) \nabla \nu(Z)^\top \frac{1}{\epsilon} \right], \tag{54}$$

*when $Z$ follows a Cauchy distribution.*

*Proof.* Proceeding in a similar way to 5, for a Cauchy distribution we have $\nu(z) = \ln(1 + z^2)$, thus

$$\mathbb{E}_Z \left[ y^*(\theta) \nabla \nu(Z)^\top \right] = y^*(\theta) \mathbb{E}_Z \left[ \frac{2Z^\top}{1 + \|Z\|^2} \right] \tag{55}$$

$$= 0 \tag{56}$$

and finally we have:

$$J_\theta y_\epsilon^*(\theta) = \mathbb{E}_Z \left[ \left( y^*(\theta + \epsilon Z) - y^*(\theta) \right) \nabla \nu(Z)^\top \frac{1}{\epsilon} \right], \tag{57}$$

for $Z$ following a Cauchy distribution $\square$

### A.3 Adaptive smoothing

In the same way as we proceeded for Jacobians computation, control variates method can be used to reduce the variance of Monte-Carlo estimators of the sensitivity of perturbed optimizers with respect to smoothing parameter $\epsilon$.

**Proposition 7.** *The sensitivity of the perturbed optimizer with respect to the smoothing parameter $\epsilon$ can be expressed as:*

$$J_\epsilon y_\epsilon^*(\theta) = \mathbb{E}_Z \left[ \left( y^*(\theta + \epsilon Z) - y^*(\theta) \right) \frac{\nabla \nu(Z)^\top Z - 1}{\epsilon} \right] \tag{58}$$

*for $Z$ a smoothing noise following a Gaussian prior.*

*Proof.* In a similar way to 5, using the same change of variable, we get:

$$J_\theta y_\epsilon^*(\theta) = \frac{\partial}{\partial \epsilon} \left( \int_{-\infty}^{\infty} y^*(\theta + \epsilon z) \mu(z) \mathrm{d}z \right), \tag{59}$$

$$= \int_{-\infty}^{\infty} y^*(z') \frac{\partial}{\partial \epsilon} \mu\left( \frac{z' - \theta}{\epsilon} \right) \mathrm{d}z', \tag{60}$$

$$= \int_{-\infty}^{\infty} y^*(z') \frac{1}{\epsilon^2} \left( \nabla \nu \left( \frac{z' - \theta}{\epsilon} \right)^\top \frac{z' - \theta}{\epsilon} - 1 \right) \mu\left( \frac{z' - \theta}{\epsilon} \right) \mathrm{d}z' \tag{61}$$

which yields with the inverse change of variable:

$$J_\epsilon y_\epsilon^*(\theta) = \mathbb{E}_Z \left[ y^*(\theta + \epsilon Z) \frac{\nabla \nu(Z)^\top Z - 1}{\epsilon} \right] \tag{62}$$

Then, for $Z$ following a Gaussian distribution, we have:

$$\mathbb{E}_Z \left[ y^*(\theta) \frac{\nabla \nu(Z)^\top Z - 1}{\epsilon} \right] = y^*(\theta) \mathbb{E}_Z \left[ \frac{\|Z\|^2 - 1}{\epsilon} \right] \tag{63}$$

$$= 0 \tag{64}$$

for $Z$ following a standard Gaussian noise. Finally, we can write:

$$J_\epsilon y_\epsilon^*(\theta) = \mathbb{E}_Z \left[ (y^*(\theta + \epsilon Z) - y^*(\theta)) \frac{\nabla \nu(Z)^\top Z - 1}{\epsilon} \right] \tag{65}$$

$\square$

**Proposition 8.** *The sensitivity of the perturbed optimizer with respect to the smoothing parameter $\epsilon$ can be expressed as:*

$$J_\epsilon y_\epsilon^*(\theta) = \mathbb{E}_Z \left[ (y^*(\theta + \epsilon Z) - y^*(\theta)) \frac{\nabla \nu(Z)^\top Z - 1}{\epsilon} \right] \tag{66}$$

*for $Z$ a smoothing noise following a Cauchy prior.*

*Proof.* We adapt the previous proof to the case of a Cauchy distribution for the noise. We observe that:

$$J_\epsilon y_\epsilon^*(\theta) = \mathbb{E}_Z \left[ y^*(\theta + \epsilon Z) \frac{\nabla \nu(Z)^\top Z - 1}{\epsilon} \right] \tag{67}$$

remains valid. In addition, we have :

$$\mathbb{E}_Z \left[ y^*(\theta) \frac{\nabla \nu(Z)^\top Z - 1}{\epsilon} \right] = y^*(\theta) \frac{1}{\epsilon} \mathbb{E}_Z \left[ \frac{2\|Z\|^2}{1 + \|Z\|^2} - 1 \right] \tag{68}$$

$$\tag{69}$$

for Z following a Cauchy noise. Moreover, an integration by parts gives:

$$\int_{-\infty}^{\infty} \frac{2x^2}{(1 + x^2)^2} \mathrm{d}x = \left[ \frac{-x}{1 + x^2} \right]_{-\infty}^{\infty} + \int_{-\infty}^{\infty} \frac{1}{1 + x^2} \mathrm{d}x \tag{70}$$

$$= [\arctan(x)]]_{-\infty}^{\infty} \tag{71}$$

so we have :

$$\mathbb{E}_Z \left[ \frac{2\|Z\|^2}{1 + \|Z\|^2} - 1 \right] = 0 \tag{72}$$

As previously done in Proposition 7, we finally get:

$$J_\epsilon y_\epsilon^*(\theta) = \mathbb{E}_Z \left[ (y^*(\theta + \epsilon Z) - y^*(\theta)) \frac{\nabla \nu(Z)^\top Z - 1}{\epsilon} \right] \tag{73}$$

$\square$

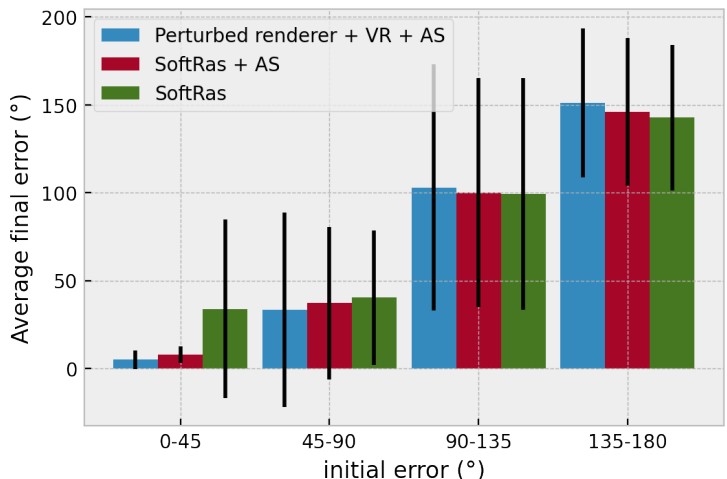

Figure 9: Pose optimization with an initial guess uniformly sampled on the rotation space. Results are reported for our differentiable perturbed renderer with Adaptive Smoothing (AS) and Variance Reduction (VR), and for SoftRas [23] with and without Adaptive Smoothing.

## B    Experimental results

In this section, we provide details on our experimental setup and some additional results. In addition, we provide the code for our differentiable perturbed renderer based on Pytorch3d [31] and the experiments on pose optimization while additional experiments will be made available upon publication.

### B.1    Pose optimization

Trying to solve pose optimization from an initial guess taken uniformly in the rotation space leads to very variable results when measuring the average final error (Fig.9). When the initialization is too far from the initial guess, the optimization process converges towards a local optima. This limitation is inherent to approaches using differentiable rendering to solve pose optimization tasks and could be avoided only by using a combination with other methods, e.g. Render & Compare [18], to get better initial guess for the pose. For this reason, we prefer to consider the ability of our differentiable renderer to retrieve the true pose with an initial guess obtained by perturbing the target and measure the amount of final errors below a given precision threshold (Fig. 10,11).

In addition to the pose optimization task on the cube, we propose to solve the same problem with others objects from Shapenet dataset [7] (Fig. 11).

### B.2    Mesh reconstruction

**Reconstruction network**    Network architecture is based on [23] and is represented in Fig. 12. The encoder is composed of 3 convolutional layers (each of them is followed by a batch normalization operations) and 3 fully connected layers. We use two different decoders to retrieve respectively the vertices' position and their color. The positional decoder is composed of 3 fully connected layers. The decoder used for colors retrieval is composed of a branch selecting a palette of $N_p$ colors on the input image while the second branch mixes these colors to colorize each of the $N_v$ vertices.

**Training setup**    We use the same setup as described in [8, 16, 23]. The training is done by minimizing a loss combining silhouette and color information with a regularization term:

$$\mathcal{L} = \lambda_{sil}\mathcal{L}_{sil} + \lambda_{RGB}\mathcal{L}_{RGB} + \lambda_{lap}\mathcal{L}_{lap}, \tag{74}$$

and we set $\lambda_{sil} = 1$, $\lambda_{RGB} = 1$, $\lambda_{lap} = 3.10^{-3}$ during training.

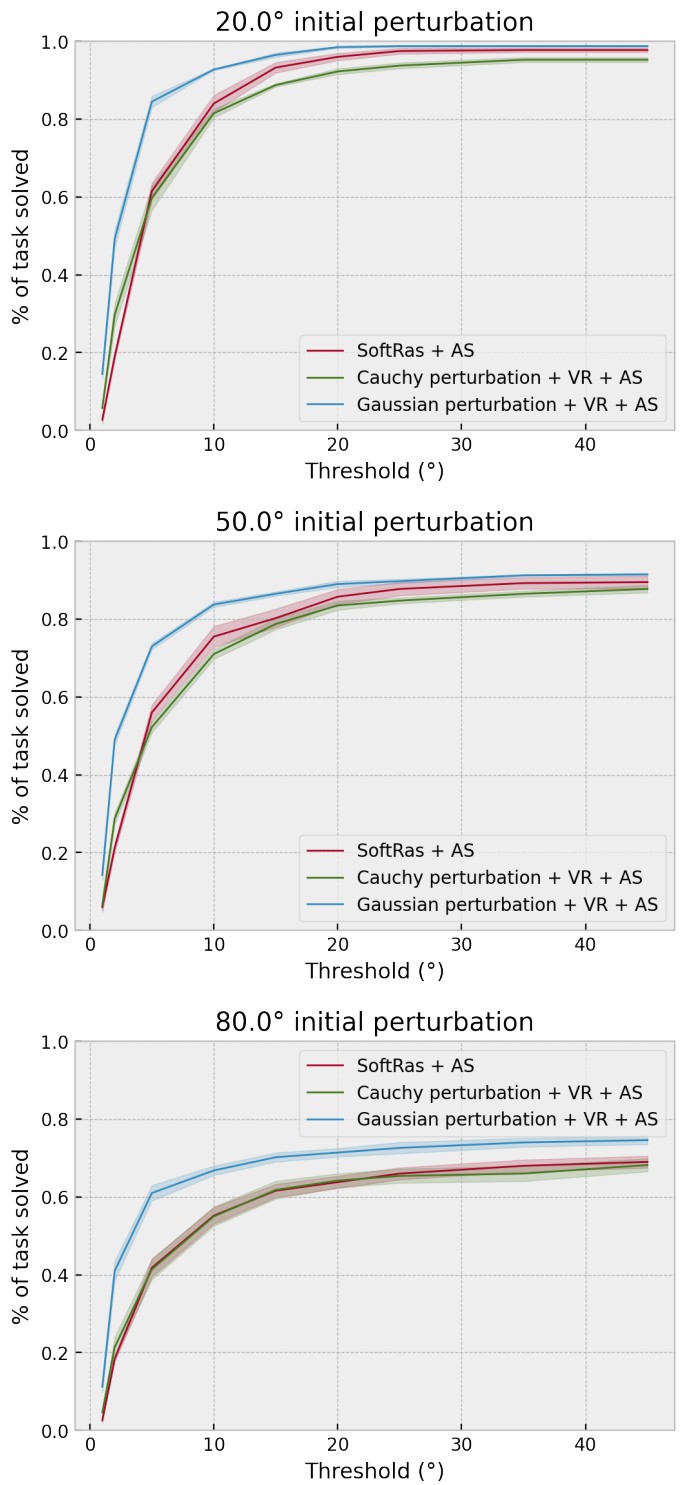

Figure 10: Pose optimization results for a perturbed renderer with Gaussian and Cauchy priors on the smoothing noise, and for SoftRas.

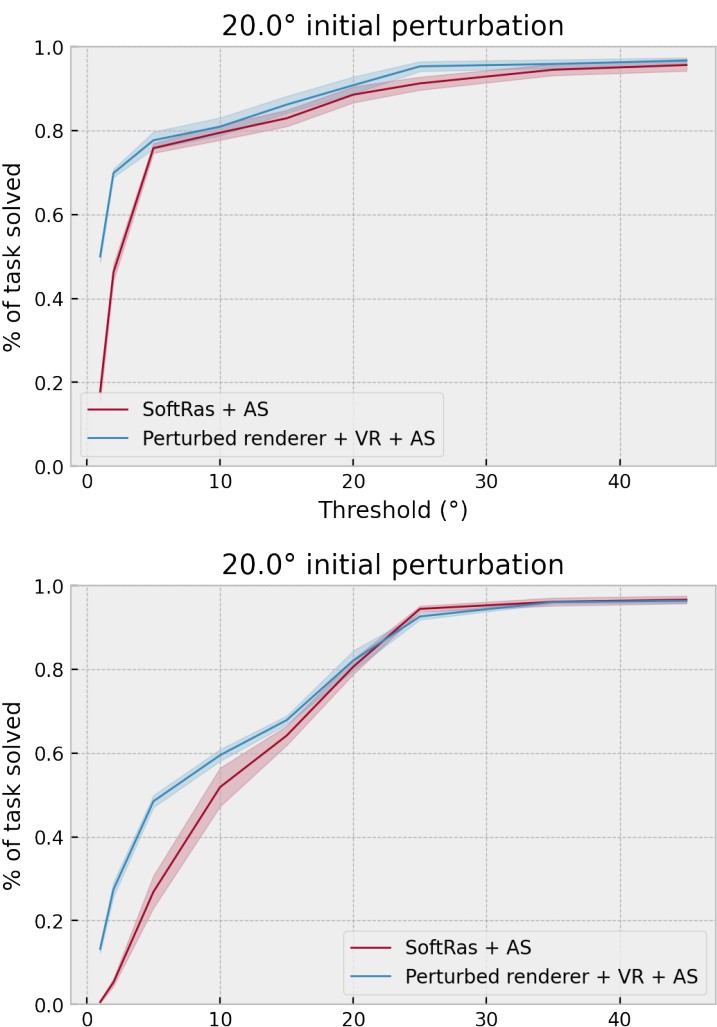

Figure 11: Pose optimization on various objects: a bag (**Top**) and a mailbox (**Bottom**)

In more details, $\mathcal{L}_{sil}$ is the negative IoU between silhouettes $I$ and $\hat{I}$ which can be expressed as:

$$\mathcal{L}_{sil} = \mathbb{E}\left[1 - \frac{\|I \odot \hat{I}\|_1}{\|I + \hat{I} - I \odot \hat{I}\|_1}\right] \tag{75}$$

$\mathcal{L}_{RGB}$ is the $\ell_1$ RGB loss between $R$ and $\hat{R}$:

$$\mathcal{L}_{RGB} = \|R - \hat{R}\|_1 \tag{76}$$

$\mathcal{L}_{lap}$ is a Laplacian regularization term penalizing the relative change in position between a vertices $v$ and its neighbours $\mathcal{N}(v)$ which can be written:

$$\mathcal{L}_{lap} = \sum_v \left(v - \frac{1}{|\mathcal{N}(v)|} \sum_{v' \in \mathcal{N}(v)} v'\right)^2 \tag{77}$$

**Additional results** We provide additional qualitative results for colorized mesh reconstruction (Fig. 13).

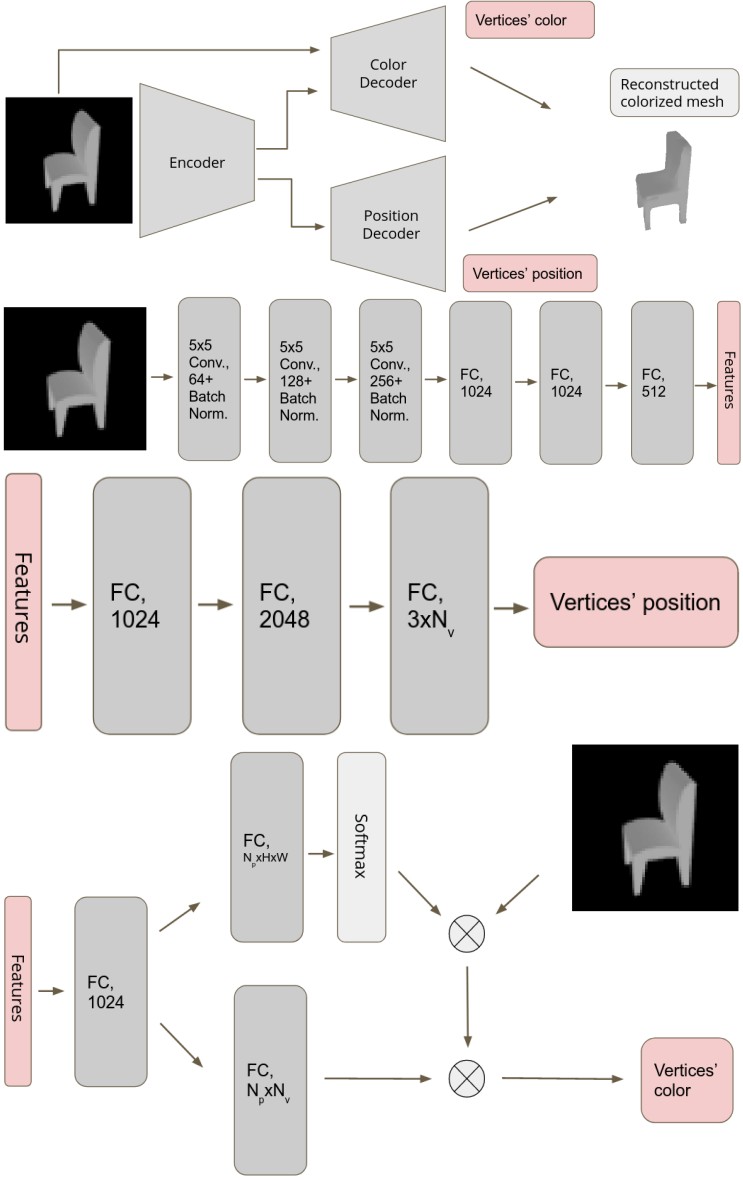

Figure 12: Learning architecture for the task of colorized 3D mesh reconstruction. **First row**: Overview of reconstruction network. **Second row**: Encoder network. **Third row**: Positional decoder. **Fourth row**: Color decoder.

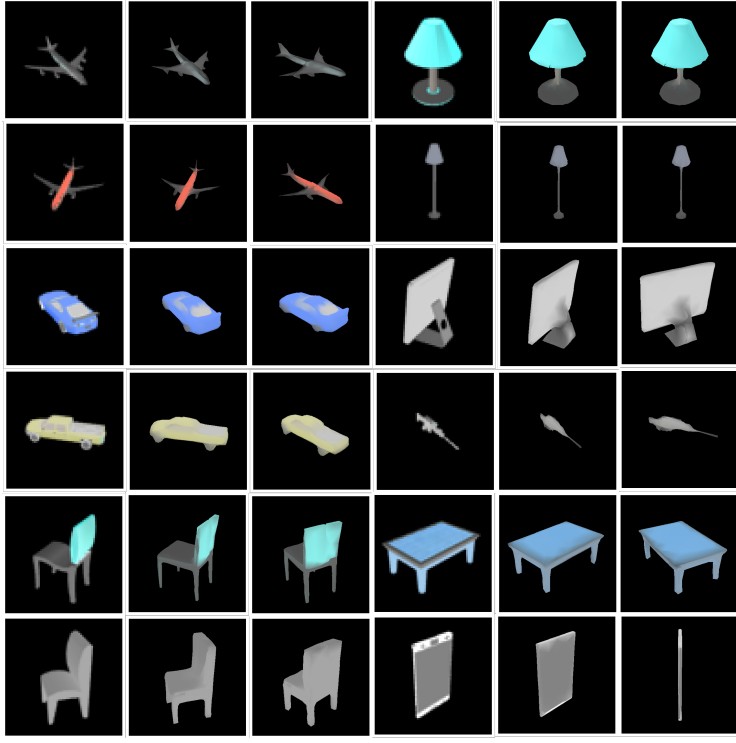

Figure 13: Results from unsupervised 3D mesh and color reconstruction. $1^{st}$ and $4^{th}$ column: Input image. Other columns: Reconstructed 3D mesh from different angles.