# OpenReview forum: "Differentiable rendering with perturbed optimizers"
_NeurIPS.cc/2021/Conference — NeurIPS 2021 Poster_

### Official Review · Reviewer_zsMR · 2021-07-13

**Rating:** 7
**Confidence:** 5

**Summary:**

This paper proposes a new differentiable rendering method, differentiable perturbed renderers,  wherein the gradients of the rasterization step are approximated using perturbed optimizers. They implement this change on top of the SoftRas method and demonstrate that it leads to better pose estimation, and on par performance with state of the art methods in unsupervised single image 3d object reconstruction.

**Ethical Concerns:**

No.

**Limitations And Societal Impact:**

Yes.

**Main Review:**

This is a strong paper. It is written clearly, and the method is explained properly. Perturbed optimizers seem to be a new way for tackling optimization over discrete operations in deep learning, and their application to rasterization is well motivated and appropriate. The theory required for applying them to this new setting seems right, and is well explained. The results are also adequate for the claims made. It would have been nice to see higher performance on the reconstruction task, though diff rendering is a relatively new and so highlighting new avenues to explore , such as your proposed perturbed optimizers,  is important, and so I do not believe it is necessary.

 I would say one nice set of experiments to see would be optimization over the same set of parameters as in the DIB-R  paper Figure 2. ( camera position, texture , vertex positions, etc. ), with qualitative and quantitative comparisons to DIB-R and SoftRas. Having this would go a long was to increasing my score, and I believe would sell you paper quite a bit better. The pose estimation seems quite a bit easier than optimizing for texture and vertex position for example, and it would be nice to see how you method fairs, even if the performance is only on par or worse.

**Time Spent Reviewing:**

3

---

> ### Author Response · Authors · 2021-08-10
> **Answer to Reviewer zsMR**
>
> We thank the reviewer for the positive feedback on our work and we are glad to see that they see a potential positive impact on the field of differentiable rendering. Moreover, we find the suggested set of experiments to be a very nice illustration of the capacities of the perturbed differentiable renderer.
> As requested by the reviewer, we performed a sanity check similar to Fig. 2 of [8] by running optimization on parameters such as camera position, lights, vertex positions or textures for simple objects. We will add a figure on this and leave for future work the extensive quantitative analysis on these tasks.
>
> References:
>
> [8] W. Chen, H. Ling, J. Gao, E. Smith, J. Lehtinen, A. Jacobson, and S. Fidler.  Learning to predict 3d objects with an interpolation-based differentiable renderer. In H. Wallach, H. Larochelle, A. Beygelzimer, F. d'Alché-Buc, E. Fox, and R. Garnett, editors,Advances in Neural Information Processing Systems, volume 32. Curran Associates, Inc., 2019

---

### Official Review · Reviewer_wBDa · 2021-07-15

**Rating:** 6
**Confidence:** 4

**Summary:**

This paper proposes a new differentiable renderer by introducing perturbed optimizers. The paper re-writes the non-differentiable rasterization and aggregation steps to heaven side functions and relaxes them to be differentiable by adding noise.
The author shows that previous work, especially softras, can be treated as a special case of the perturbed renderer with specific prior selections. The authors also demonstrated in several experiments they achieve better or comparable results than the state of the art.

**Ethics Review Area:**

["I don’t know"]

**Limitations And Societal Impact:**

There are several weak points of this paper.

1) The paper is not self-contained.  I have a hard time understanding each equation. While the authors largely rely on [4], it makes readers hard to follow without knowing [4]. Moreover, the notations in this paper are very vague.   E.g. line 66 it should be j \in [1,.., m]?  line 76 z should be depth order?  If it is depth, how do you compute <z, y>?  line 88, what's v?  The paper writing definitely needs to be improved.


2) Relation of softras and noise types.  The authors show that softras can be treated as the special case of the perturbed renderer. Softras has some hyperparameters which are hard to explain but in the new renderer, all the parameters have clear noise intensity meaning.  While this is very good, but the paper lacks further analysis of which noise should we use. E.g. in table 1, it seems Gaussian noise is the best. But could it be generalized to other cases?


3) Comparison.  In line 214 the authors said, "Unless stated, the results for the other renderers are extracted from their original publications, as we were not able to get access to their implementations."  I am a little confused about this.  Clearly, N3mr, softras, dibr released their renderers. Thus, why not simply replace the renderer from perturbed renderer to new renderers and compare the score?  Directly copying scores from others' papers may have the wrong comparison. While all the methods use shapenet, the rendered images could vary from lighting to camera views. I would highly recommend the authors redo the comparison.


4)  The authors assume each pixel will be influenced by all the triangles. For each pixel, it renders the influences of all the triangles and combines them together(Eq 3).  I wonder what's the rendering speed of the perturbed renderer? Theoretically, such design plus the MC sampling will make the renderer very time-consuming.  The authors should show the rendering speed comparison.


5) Following the last concern.  Generally speaking, rasterization accelerates the rendering speed by assuming each pixel is influenced by the frontest triangle.  On the other side, MC is always used in ray tracing to compute multiple bounces. The authors introduce MC in rasterization, which seems to adopt the weak points of two rendering methods.   I wonder did the authors consider extending it to RT?

Post rebuttal:

The authors addressed my concerns and I will raise the score to be marginally above the threshold and be happy to see this paper get accepted.








**Main Review:**

Strengths

1) Differentiable reformulation
This paper treats rasterization as a linear programing problem.  It extends the [4] to the rendering pipeline and relaxes the non-differentiable rendering to be differentiable by introducing perturbed optimizers. It provides a new differentiable renderer from a new perspective.

2) Better performance
The authors show AS and VR are very useful in optimization, for both perturbed renderer and softras. In the 6D pose comparison, the results are better than softras.




**Time Spent Reviewing:**

4

---

> ### Author Response · Authors · 2021-08-10
> **Answer to Reviewer wBDa**
>
> We thank the reviewer for providing insightful comments to our work, which we address below.
>
> > The paper writing definitely needs to be improved.
>
> We thank the reviewer for the thorough reading. Indeed, "n" should be replaced by "m" on L.66 and we should precise that $\nu$ is a generic differentiable function. We apologize for this misnotation and will modify the paper accordingly. $z_j$ is the inverse depth of the $j^{th}$ triangle as it is stated on L.76. As detailed on L.74-75, the argmax operation consists in selecting the active pixel with the largest $z$ which corresponds to the Z-Buffer operation in classical renderers and, thus, the dot product is never directly computed. We are aware that the theory in [4] may be hard to understand, hence, we have made an attempt to provide a short intuitive explanation on L.80-95 and in Fig. 1. We will improve the explanation of [4] by adding insights on equations (5) and (6) drawing links to finite differences and random optimization theory [b].
>
> > Relation of softras and noise types. The authors show that softras can be treated as the special case of the perturbed renderer. Softras has some hyperparameters which are hard to explain but in the new renderer, all the parameters have clear noise intensity meaning. While this is very good, but the paper lacks further analysis of which noise should we use. E.g. in table 1, it seems Gaussian noise is the best. But could it be generalized to other cases?
>
> As shown by our experiments, Gaussian prior appears to perform better than Cauchy and Gumbel prior (see Tab. 1 and Fig.6 of the paper and Fig. 3 of appendix). Qualitatively, the Gumbel distribution is not symmetric and there is no reason to privilege one direction rather than another during smoothing. This symmetry  of the Gaussian kernel can explain better results obtained with the Gaussian smoothing. We have also explored the use of Cauchy distribution as it is heavy-tailed which naturally increases the smoothing and allows to get stronger gradients. However, a too strong smoothing can also deteriorate the local information and lead to gradients weakly correlated to the original problem. This phenomenon would be the cause of worse performance of the Cauchy prior on both pose optimization and mesh reconstruction tasks. We thank the reviewer for starting this discussion that we find very interesting and we will add it to the experimental section. We provide a short discussion on priors on L.42-63 of the appendix. In particular, we point out that noise with finite support distributions would lead to some existing differentiable renderers with local gradients. The investigation of more advanced priors will be an interesting direction for future work.
>
> > In line 214 the authors said, "Unless stated, the results for the other renderers are extracted from their original publications, as we were not able to get access to their implementations." [...] Directly copying scores from others' papers may have the wrong comparison. While all the methods use shapenet, the rendered images could vary from lighting to camera views. I would highly recommend the authors redo the comparison.
>
> As mentioned by the reviewer, the implementations of the differentiable renderers SoftRas or DIB-R are available. However, their code for pose optimization and colorized mesh reconstruction was not released which makes the results difficult to reproduce. For the mesh reconstruction task, the datasets, cameras and lighting conditions were fixed in the original code of NMR. Still, as suggested by the reviewer, we re-run experiments and obtained the following results:
>
>
> |        |      Airplane     |      Bench     |     Dresser     |       Car     |      Chair     |   Display |  Lamp   |
> |--------|:----------:|:---------:|:---------:|:---------:|:-------------------:|: ----------:|:-----------|
> | Ours      |   63.5 |   49.4 | 67.1 | 72.3 |  51.6  | 60.4 |  44.3    |
> | DIB-R     |   59.7 | 50.9 | 66.2 |  72.6 | 52.0 | 57.9 | 43.8  |
> | SoftRas     |  62.0 |  47.55 | 66.2 | 69.4  | 49.4  | 60.0 | 43.3   |
>
> |       |      Speaker     |      Rifle     |     Sofa     |       Table     |      Phone     |   Vessel |  Mean   |
> |-----|:----------:|:---------:|:---------:|:---------:|:-------------------:|: ----------:|:-----------|
> | Ours  |   62.8 |   65.7 | 66.7 | 49.2 |  80.2   |  60.0 |    61.0      |
> | DIB-R  |   63.8 | 61.0 | 65.4 |  50.5 | 76.3 | 58.6 | 59.9  |
> | SoftRas  | 62.6 |  61.4 | 60.4 |  43.6 |  76.4  |  59.9 | 58.6 |
>
>
> As mentioned in the general comment:
>
> *From the results above we observe that our method outperforms DIB-R and SoftRas when run in comparable settings. In particular, the large gap in performance for the Dresser and Car categories observed in Table 2 has been removed. These results together with results for the pose estimation task in Table 1 clearly demonstrate the practical benefits of our approach.*
>
> In addition, we believe that pose optimization and mesh reconstruction tasks represent good benchmarks to evaluate differentiable rendering techniques, so we will release our implementation (compatible with SoftRas, DIB-R and our perturbed renderer) to facilitate future research in this area.
>
> > For each pixel, it renders the influences of all the triangles and combines them together(Eq 3). I wonder what's the rendering speed of the perturbed renderer? Theoretically, such design plus the MC sampling will make the renderer very time-consuming. The authors should show the rendering speed comparison.
>
> As mentioned in the general comment:
>
> *We agree that the complexity of MC estimators grows linearly with the number of samples.  In our work we propose the adaptive smoothing (AS) and variance reduction (VR) schemes that efficiently reduce the computational burden of our perturbed renderer (see Fig. 5). As shown in Fig. 5 (Left), the performance of our perturbed renderer is already high when using as little as two samples, and typically saturates when using around 10 samples at each iteration.*
>
> *Given the limited number of required samples, the time complexity of our method can be kept constant using parallelization (see L.184-185). To validate this claim, we have measured the runtime of our method averaged over 100 forward and backward optimization steps using Nvidia RTX6000 GPU. As shown by the following table, the computation time of our method does not depend on the number of samples and is comparable to Softras.*
>
> | # of samples   |   1    |   2  |   8  |  64  |  SoftRas | Hard Renderer |
> |----------------|:----------:|:---------:|:---------:|:---------:|:-------------------:|: ----------:|
> | Forward (ms)  |  30 (+/- 3) |   30 (+/- 3) | 31 (+/- 3) | 31 (+/- 3) |     29(+/- 3)   | 29 (+/- 3) |
> | Backward (ms)  | 19 (+/- 1) | 19  (+/- 1) | 19 (+/- 2) |  22 (+/- 1 ) | 18    (+/- 1)  | N/A |
> | Total runtime (s)   |  4.93 (+/- 0.18) |  4.87 (+/- 0.07) | 5.00 (+/- 0.35) |  5.35 (+/- 0.12) |   4.68   (+/- 0.19)  | N/A |
>
> Concerning the cost of considering every triangle for every pixel it is true that it makes the backward computation more costly by backpropagating non-null gradients. However, this choice was introduced in SoftRas [22] and leads to a differentiable rendering pipeline by providing gradients even for faces that are far from their "target" position. Our implementation relies on Pytorch3d [29] and thus also offers the possibility to only consider triangles which are the K-nearest to a pixel in terms of depth (z) or distance in the xy-plane. In addition, the modularity of our approach makes it possible to combine a soft rasterization and a hard aggregation step to backpropagate gradients only in the xy-plane to foreground faces. This modularity offers a trade-off to the user who can choose to put some gradients to zero for reducing the computation time (see Fig. 4 in [29]).
>
> > Generally speaking, rasterization accelerates the rendering speed by assuming each pixel is influenced by the frontest triangle. On the other side, MC is always used in ray tracing to compute multiple bounces. The authors introduce MC in rasterization, which seems to adopt the weak points of two rendering methods. I wonder did the authors consider extending it to RT?
>
> In the case of ray tracing, MC is used to evaluate the rendering equation [c] and proceeds by casting several rays, each of them making several bounces. In our case, we use MC estimators to evaluate several hard rasterizers and aggregaters in parallel (see Fig. 1), an operation which is  much cheaper than ray casting [15]. Thus, MC estimators used in our work and in ray tracing are not comparable as they evaluate operations of very different computational cost. This is the reason why our differentiable renderer has computation times similar to rasterization based renderers (see statistics in the general comment). Nonetheless, we believe that applying the perturbed optimizers technique in the ray tracing case is an interesting topic. The perturbed optimizers are originally designed to smooth and provide gradients for discrete argmax operations. From our understanding, it does not seem that such operations are involved in the ray tracing process which makes it difficult to think about a direct application.
>
> References:
>
> [15]  H. Kato, Y. Ushiku, and T. Harada. Neural 3d mesh renderer, 2017
>
> [22] S. Liu, T. Li, W. Chen, and H. Li. Soft rasterizer: A differentiable renderer for image-based 3d reasoning,2019.
>
> [29] N. Ravi, J. Reizenstein, D. Novotny, T. Gordon, W.-Y. Lo, J. Johnson, and G. Gkioxari. Accelerating 3d deep learning with pytorch3d. arXiv:2007.08501, 2020
>
> [b] Y. Nesterov and V. Spokoiny. Random gradient-free minimization of convex functions. Foundations of Computational Mathematics, 17(2):527–566, 2017.
>
> [c] J. T. Kajiya. The rendering equation. In Proceedings of the 13th annual conference on Computer graphics and interactive techniques, pages 143–150, 1986

---

> > ### Comment · Reviewer_wBDa · 2021-08-26
> > **Thanks for the rebuttal**
> >
> > Thanks for the detailed rebuttal.  My major concerns are 1) explanation of noise  2) comparison 3) running time. The authors address all of them in the rebuttal. For the noise, I am satisfied if the authors add more discussions in the paper. I agree further study of noise can be left in future work.  For the comparison, the authors show their method outperforms softras and dibr in the same setting. For the running time, it seems the proposed method runs as fast as softras.  I would encourage the authors to evaluate the memory cost as well, which should be easy to add in the final version.  While the proposed method didn't show any new capacity compared to previous methods, I think an alternative new differentiable method is valuable. Thus, I will raise my score to marginally above the threshold.

---

> > > ### Author Response · Authors · 2021-09-01
> > > **Answer to Reviewer wBDa**
> > >
> > > Thank you for your response and your suggestion.
> > > >  I would encourage the authors to evaluate the memory cost as well, which should be easy to add in the final version.
> > >
> > > We have done a similar experiment as for the runtime comparison and measured the maximal allocated memory:
> > >
> > > | # of samples   |      1     |      2     |   4    |    8     |  16    |     32      |       64       |    128        |    256  |      SoftRas     |   Hard Renderer |
> > > |----------------|:----------:|:---------:|:---------:|:---------:|:-------------------:|: ----------:|: ----------:|: ----------:|: ----------:|: ----------:|: ----------:|
> > > | max memory (Mb)          |  217.6 |   217.6 | 217.6| 217.6 |     236.0   |    303.1     |    440.0    |     704.6     |    1241.1     |    180.3  | 178.5 |
> > >
> > > As expected, the memory grows linearly with the number of samples N for N>8. For N<=8 the memory footprint is constant as the estimator does not represent the dominant cost in memory. Moreover, as we show in Fig.5 (8 MC samples), using eight samples already results in a high performance, hence, the memory consumption of our method is similar to the one of SoftRas and can be handled by the current GPUs (we used Nvidia Tesla V100 and RTX 6000 GPUs which both have around 20 Gb memory capacity). We find this analysis to be interesting and will add it to the final version of the paper as suggested by the reviewer.

---

### Official Review · Reviewer_yvH4 · 2021-07-16

**Rating:** 4
**Confidence:** 3

**Summary:**

This paper proposes a differentiable rendering method that builds connection between the 3D model and 2D perspective appearance. Different from previous methods, the differentiable rendering is formulated into an optimization problem. Two computer vision tasks, including pose estimation and 3D shape from color image, are evaluated with the proposed renderer and compared to SoftRas [22].

**Limitations And Societal Impact:**

No comment.

**Main Review:**

Strength:
- The performance for pose estimation outperforms existing methods.
- The formulation seems novel.

Weakness:
- Could the proposed formulation be applied to other 3D shape representations, e.g. point cloud, implicit? If so, an experiment should be added. If not, then better to mention this is a mesh-specific technic.

- The algorithm seems to be computational costy, e.g. 1 min per image. Would this heavily affect the training efficiency? While comparing to other methods, e.g. SoftRas, the run-time efficiency and hyper-parameters, e.g. number of iteration, training epoch, should be mentioned. Otherwise the comparison may not be on a fair basis.

- Keeping the boundary differentiable is one of the main issues in existing differentiable rendering technics, which is also argued mainly in this paper. However it is not clear how this is improved by the proposed method. From Fig 2 and 4, it seems the boundary area has obvious artifacts. Also some experiments should be performed to highlight the improvements over occlusion boundary.

- The performance on 3D shape prediction does not outperform existing approaches.

Overall, despite a novel formulation, the improvements over existing approaches are not clear. As shown in the experiment, the propose method does not necessarily produce more accurate rendering or support better for vision tasks. Considering that the proposed tech is also quite slow, the overall advantage over existing approaches is not very clear.

**Time Spent Reviewing:**

3

---

> ### Author Response · Authors · 2021-08-10
> **Answer to Reviewer yvH4**
>
> We thank the reviewer for the comments and try to answer the raised questions and possible misunderstandings below.
>
> > Could the proposed formulation be applied to other 3D shape representations, e.g. point cloud, implicit? If so, an experiment should be added. If not, then better to mention this is a mesh-specific technic.
>
> Thank you for this suggestion of possible extension for our work. Indeed, the same techniques can be applied to render point clouds in a differentiable way. As detailed in [29], challenges to overcome when rendering point clouds are identical to the ones encountered with meshes as non-differentiabilities emerge because of rasterization and depth-aggregation operations. Thus, implementing a differentiable perturbed renderer for point clouds would be a direct extension of our work.
>
> Regarding implicit shape representations, implicit shape rendering is a naturally differentiable operation because the occupancy function is differentiable in general [a] (this is also true for the transmission function) which differs from the cases of meshes or point clouds whose structures are inherently discrete. It is exactly these discrete structures that make rasterization and aggregation steps non differentiable and requires the use of perturbed optimizers (see lines 78-79). For this reason, the perturbed optimizers proposed in our paper are not directly appropriate for implicit shape rendering.
>
> > The algorithm seems to be computational costy, e.g. 1 min per image. Would this heavily affect the training efficiency? While comparing to other methods, e.g. SoftRas, the run-time efficiency and hyper-parameters, e.g. number of iteration, training epoch, should be mentioned. Otherwise the comparison may not be on a fair basis.
>
> We refer to the general comment for more details on this and we repeat it here:
>
> *We agree that the complexity of MC estimators grows linearly with the number of samples.  In our work we propose the adaptive smoothing (AS) and variance reduction (VR) schemes that efficiently reduce the computational burden of our perturbed renderer (see Fig. 5). As shown in Fig. 5 (Left), the performance of our perturbed renderer is already high when using as little as two samples, and typically saturates when using around 10 samples at each iteration.*
>
> *Given the limited number of required samples, the time complexity of our method can be kept constant using parallelization (see L.184-185). To validate this claim, we have measured the runtime of our method averaged over 100 forward and backward optimization steps using Nvidia RTX6000 GPU. As shown by the following table, the computation time of our method does not depend on the number of samples and is comparable to Softras.*
>
> | # of samples   |      1     |      2     |       8     |       64     |      SoftRas     |   Hard Renderer |
> |----------------|:----------:|:---------:|:---------:|:---------:|:-------------------:|: ----------:|
> | Forward (ms)          |    30 (+/- 3) |   30 (+/- 3) | 31 (+/- 3) | 31 (+/- 3) |     29(+/- 3)   | 29 (+/- 3) |
> | Backward (ms)          |   19 (+/- 1) | 19  (+/- 1) | 19 (+/- 2) |  22 (+/- 1 ) | 18    (+/- 1)  | N/A |
> | Total runtime (s)          |  4.93 (+/- 0.18) |  4.87 (+/- 0.07) | 5.00 (+/- 0.35) |  5.35 (+/- 0.12) |   4.68   (+/- 0.19)  | N/A |
>
> *We will add the above results and discussion to the final version of the paper, if accepted.*
>
> Moreover, we believe there might be a misunderstanding regarding "1 min per image". 1 minute in our experiments is an approximate time required to solve one pose optimization task which includes hundreds of rendering steps. The rendering of a single image, on the other hand, takes only about 30ms with our method.
>
>
> > Keeping the boundary differentiable is one of the main issues in existing differentiable rendering technics, which is also argued mainly in this paper. However it is not clear how this is improved by the proposed method. From Fig 2 and 4, it seems the boundary area has obvious artifacts. Also some experiments should be performed to highlight the improvements over occlusion boundary.
>
> We believe that by "artefacts" the reviewer may refer to the blur and transparency effects of the differentiable perturbed renderer as visible e.g. in Figure 2. To clarify this comment, we would like to emphasize that the main purpose of our proposed differentiable renderer is to obtain gradients with respect to scene parameters that should further enable to solve subsequent tasks such as pose optimization and mesh reconstruction. We further emphasize that our work does not aim to provide a photorealistic rendering technique. We agree that keeping boundaries differentiable is one of the challenges of differentiable rendering. With this in mind, the blurry and transparency effects (which are common to others differentiable renderers e.g. SoftRas [22], NMR [15], DIB-R [8]) are actually at the core of the approach as these perturbations allow to make the rasterization and aggregation steps differentiable (see Fig. 1 for illustration).
>
> > The performance on 3D shape prediction does not outperform existing approaches.
>
> We refer to the general comment for more details on this point and we repeat it here:
>
> *We note that results for other methods reported for the shape reconstruction task in Table 2 are obtained from original papers and may use slightly different experimental setups compared to ours. Following the suggestion of Reviewer wBDa, we use original SoftRas and DIB-R implementations with their default hyper-parameters and re-run  the colorized mesh reconstruction experiments using the same scene parameters for all tested methods.*
>
> |        |      Airplane     |      Bench     |     Dresser     |       Car     |      Chair     |   Display |  Lamp   |
> |--------|:----------:|:---------:|:---------:|:---------:|:-------------------:|: ----------:|:-----------|
> | Ours           |   63.5 |   49.4 | 67.1 | 72.3 |  51.6  | 60.4 |  44.3    |
> | DIB-R          |   59.7 | 50.9 | 66.2 |  72.6 | 52.0 | 57.9 | 43.8  |
> | SoftRas     |  62.0 |  47.55 | 66.2 | 69.4  | 49.4  | 60.0 | 43.3   |
>
> |       |      Speaker     |      Rifle     |     Sofa     |       Table     |      Phone     |   Vessel |  Mean   |
> |-----|:----------:|:---------:|:---------:|:---------:|:-------------------:|: ----------:|:-----------|
> | Ours           |   62.8 |   65.7 | 66.7 | 49.2 |  80.2   |  60.0 |    61.0      |
> | DIB-R          |   63.8 | 61.0 | 65.4 |  50.5 | 76.3 | 58.6 | 59.9  |
> | SoftRas     | 62.6 |  61.4 | 60.4 |  43.6 |  76.4  |  59.9 | 58.6 |
>
> *From the results above we observe that our method outperforms DIB-R and SoftRas when run in comparable settings. In particular, the large gap in performance for the Dresser and Car categories observed in Table 2 has been removed. These results together with results for the pose estimation task in Table 1 clearly demonstrate the practical benefits of our approach.*
>
> References:
>
> [8] W. Chen, H. Ling, J. Gao, E. Smith, J. Lehtinen, A. Jacobson, and S. Fidler.  Learning to predict 3d objects with an interpolation-based differentiable renderer. In H. Wallach, H. Larochelle, A. Beygelzimer, F. d'Alché-Buc, E. Fox, and R. Garnett, editors,Advances in Neural Information Processing Systems, volume 32. Curran Associates, Inc., 2019
>
> [15]  H. Kato, Y. Ushiku, and T. Harada. Neural 3d mesh renderer, 2017
>
> [22] S. Liu, T. Li, W. Chen, and H. Li. Soft rasterizer: A differentiable renderer for image-based 3d reasoning,2019.
>
> [29] N. Ravi, J. Reizenstein, D. Novotny, T. Gordon, W.-Y. Lo, J. Johnson, and G. Gkioxari. Accelerating 3d deep learning with pytorch3d. arXiv preprint arXiv:2007.08501, 2020
>
> [a] B. Mildenhall, P. P. Srinivasan, M. Tancik, J. T. Barron, R. Ramamoorthi, and R. Ng. Nerf: Representing scenes as neural radiance fields for view synthesis. In European conference on computer vision, pages 405–421. Springer, 2020.

---

### Official Review · Reviewer_qAqf · 2021-07-16

**Rating:** 6
**Confidence:** 4

**Summary:**

This paper presents a method for converting a standard non-differentiable rasterization-based renderer into a differentiable counterpart using recent advances in the perturbed optimizer. The main contribution lies in the generic nature provided by the theoretical formulation, where some specifically chosen priors in the proposed framework can retrieve recent state-of-the-art differentiable renderers, e.g. SoftRas.
As the proposed approach mainly relies on a Monte-Carlo estimator for generating the smoothing parameters. Variance-reduction mechanism and sensitivity analysis are also incorporated into the proposed framework to ensure the robustness of the approximate renderer. Experiments on pose estimation and single-view reconstruction are conducted for comparing the performance of the proposed method and previous state-of-the-art approaches.

**Limitations And Societal Impact:**

Yes

**Main Review:**

Pros:

- By introducing the perturbed optimizer to analyzing the non-differentiability of standard rasterization-based rendering pipeline provides new insight into constructing a generic differentiable renderer. The perturbed differentiable rendering can also be implemented on top of existing renderers without major modifications, which makes this approach versatile and a potential performance booster for existing DR techniques.

Cons:

- In terms of technical novelty, it seems to be a direct application of perturbed optimizer to differentiable rendering. Though it requires some efforts (such as the variance reduction and sensitivity analysis) to adapt the existing theory to this specific application, it is not very challenging as there are existing protocols to follow.

- Evaluation-wise, in the task of single-view reconstruction, though the proposed method can achieve comparable performance compared to the state-of-the-art approaches, it still fails to outperform DIB-R in mean accuracy and also outperformed by SoftRas in many categories, such as the Dresser, Car, etc. The 3D pose estimation is only evaluated on a simple cube which cannot provide a very comprehensive measurement of the proposed approach.

**Time Spent Reviewing:**

3

---

> ### Author Response · Authors · 2021-08-10
> **Answer to Reviewer qAqf**
>
> We thank the reviewer for the feedback and we address their concerns below.
>
> > In terms of technical novelty, it seems to be a direct application of perturbed optimizer to differentiable rendering. Though it requires some efforts (such as the variance reduction and sensitivity analysis) to adapt the existing theory to this specific application, it is not very challenging as there are existing protocols to follow.
>
> We agree that the core optimization techniques used in our work are not novel. However, we believe that our formulation of differentiable rendering as perturbed instances is new, non-trivial and leads to a unified approach generalizing existing differentiable renderers. As a more principled approach, our formulation may lead to better understanding of current differentiable renderers. We would like to stress that the Variance Reduction and Adaptive Smoothing schemes are contributions that are not present in the original paper [4] and are key elements to make perturbed optimizers practically useful by significantly alleviating their computational burden (see Fig 5.). While [4] only explored the use of perturbed optimizers at final layers, embedding perturbed optimizers at intermediate layers as we do in our work is a novelty. This is only made possible with variance reduction as we want to avoid noise to propagate into the pipeline. Moreover, we believe that adaptive smoothing via sensitivity analysis is a contribution in itself as it provides a general way to adapt smoothing parameters and can be applied to improve performance of others differentiable renderers (see Fig. 5).
>
> > Evaluation-wise, in the task of single-view reconstruction, though the proposed method can achieve comparable performance compared to the state-of-the-art approaches, it still fails to outperform DIB-R in mean accuracy and also outperformed by SoftRas in many categories, such as the Dresser, Car, etc. The 3D pose estimation is only evaluated on a simple cube which cannot provide a very comprehensive measurement of the proposed approach.
>
> We refer to the general comment for more details on the mesh-reconstruction task, and we repeat it here:
>
> *We note that results for other methods reported for the shape reconstruction task in Table 2 are obtained from original papers and may use slightly different experimental setups compared to ours. Following the suggestion of Reviewer wBDa, we use original SoftRas and DIB-R implementations with their default hyper-parameters and re-run  the colorized mesh reconstruction experiments using the same scene parameters for all tested methods.*
>
> |        |      Airplane     |      Bench     |     Dresser     |       Car     |      Chair     |   Display |  Lamp   |
> |--------|:----------:|:---------:|:---------:|:---------:|:-------------------:|: ----------:|:-----------|
> | Ours           |   63.5 |   49.4 | 67.1 | 72.3 |  51.6  | 60.4 |  44.3    |
> | DIB-R          |   59.7 | 50.9 | 66.2 |  72.6 | 52.0 | 57.9 | 43.8  |
> | SoftRas     |  62.0 |  47.55 | 66.2 | 69.4  | 49.4  | 60.0 | 43.3   |
>
> |       |      Speaker     |      Rifle     |     Sofa     |       Table     |      Phone     |   Vessel |  Mean   |
> |-----|:----------:|:---------:|:---------:|:---------:|:-------------------:|: ----------:|:-----------|
> | Ours           |   62.8 |   65.7 | 66.7 | 49.2 |  80.2   |  60.0 |    61.0      |
> | DIB-R          |   63.8 | 61.0 | 65.4 |  50.5 | 76.3 | 58.6 | 59.9  |
> | SoftRas     | 62.6 |  61.4 | 60.4 |  43.6 |  76.4  |  59.9 | 58.6 |
>
> *From the results above we observe that our method outperforms DIB-R and SoftRas when run in comparable settings. In particular, the large gap in performance for the Dresser and Car categories observed in Table 2 has been removed. These results together with results for the pose estimation task in Table 1 clearly demonstrate the practical benefits of our approach.*
>
> For the pose optimization task, we compare our approach to SoftRas for several different objects. We report some results in the appendix and confirm the superiority of our method over SoftRas on this task (see Fig. 4 of appendix). We apologize for not referring to these results in the main paper and will correct this in the final version.
>
> References:
>
> [4] Q. Berthet, M. Blondel, O. Teboul, M. Cuturi, J.-P. Vert, and F. Bach. Learning with differentiable perturbed optimizers, 2020

---

### Author Response · Authors · 2021-08-10
**General comment**

We thank the reviewers for their feedback on our work. We are happy reviewers appreciate the novelty and the generality of our approach. Before addressing individual comments of each reviewer, we here answer points raised in multiple reviews.

**1. Computational complexity**

Several reviewers comment on the computational complexity of our method. We agree that the complexity of MC estimators grows linearly with the number of samples.  In our work we propose the adaptive smoothing (AS) and variance reduction (VR) schemes that efficiently reduce the computational burden of our perturbed renderer (see Fig. 5). As shown in Fig. 5 (Left), the performance of our perturbed renderer is already high when using as little as two samples, and typically saturates when using around 10 samples at each iteration.

Given the limited number of required samples, the time complexity of our method can be kept constant using parallelization (see L.184-185). To validate this claim, we have measured the runtime of our method averaged over 100 forward and backward optimization steps using Nvidia RTX6000 GPU. As shown by the following table, the computation time of our method does not depend on the number of samples and is comparable to Softras.

| # of samples   |      1     |      2     |       8     |       64     |      SoftRas     |   Hard Renderer |
|----------------|:----------:|:---------:|:---------:|:---------:|:-------------------:|: ----------:|
| Forward (ms)          |    30 (+/- 3) |   30 (+/- 3) | 31 (+/- 3) | 31 (+/- 3) |     29(+/- 3)   | 29 (+/- 3) |
| Backward (ms)          |   19 (+/- 1) | 19  (+/- 1) | 19 (+/- 2) |  22 (+/- 1 ) | 18    (+/- 1)  | N/A |
| Total runtime (s)          |  4.93 (+/- 0.18) |  4.87 (+/- 0.07) | 5.00 (+/- 0.35) |  5.35 (+/- 0.12) |   4.68   (+/- 0.19)  | N/A |

We will add the above results and discussion to the final version of the paper, if accepted.


**2. Performance on the shape reconstruction task**

Some reviewers comment on the lack of SOTA improvements for the shape reconstruction task. We note that results for other methods reported for the shape reconstruction task in Table 2 are obtained from original papers and may use slightly different experimental setups compared to ours. Following the suggestion of Reviewer wBDa, we use original SoftRas and DIB-R implementations with their default hyper-parameters and re-run  the colorized mesh reconstruction experiments using the same scene parameters for all tested methods.

|        |      Airplane     |      Bench     |     Dresser     |       Car     |      Chair     |   Display |  Lamp   |
|--------|:----------:|:---------:|:---------:|:---------:|:-------------------:|: ----------:|:-----------|
| Ours           |   63.5 |   49.4 | 67.1 | 72.3 |  51.6  | 60.4 |  44.3    |
| DIB-R          |   59.7 | 50.9 | 66.2 |  72.6 | 52.0 | 57.9 | 43.8  |
| SoftRas     |  62.0 |  47.55 | 66.2 | 69.4  | 49.4  | 60.0 | 43.3   |

|       |      Speaker     |      Rifle     |     Sofa     |       Table     |      Phone     |   Vessel |  Mean   |
|-----|:----------:|:---------:|:---------:|:---------:|:-------------------:|: ----------:|:-----------|
| Ours           |   62.8 |   65.7 | 66.7 | 49.2 |  80.2   |  60.0 |    61.0      |
| DIB-R          |   63.8 | 61.0 | 65.4 |  50.5 | 76.3 | 58.6 | 59.9  |
| SoftRas     | 62.6 |  61.4 | 60.4 |  43.6 |  76.4  |  59.9 | 58.6 |

From the results above we observe that our method outperforms DIB-R and SoftRas when run in comparable settings. In particular, the large gap in performance for the Dresser and Car categories observed in Table 2 has been removed. These results together with results for the pose estimation task in Table 1 clearly demonstrate the practical benefits of our approach. We hope our work will inspire new methods for differentiable rendering that will build on perturbed optimizers and will improve results further by, for example, considering alternatives to the Gaussian smoothing prior used in our work.

---

### Public Comment · ~Ahmed_Abbas1 · 2021-11-12
**Connection to black box backprop**

Hi,

Very interesting work! I have two questions:

1. Is your method extendable to general (I)LPs?
2. How would you connect eq.14 of your work with eq. 5 from https://openreview.net/forum?id=BkevoJSYPB (Differentiation of Blackbox Combinatorial Solvers). We built on this work in https://arxiv.org/abs/2106.03188 where we add smoothing during backward pass with improved convergence.

Thanks!
Ahmed

---

### Decision · Program_Chairs · 2021-09-27

**Decision:**

Accept (Poster)

**Comment:**

This paper has mixed reviewers. Most of reviewers agree with the novelty and contribution of proposing a new differentiable rendering method, differentiable perturbed renderers. The whole paper is solid and experiments are good enough to validate the contribution. Reviewer yvH4 had some serious concerns about the practical issues of the proposed methods, including generalization ability to other 3D shape representations, computational expensive, not competitive to existing approaches. The authours clarified some of these points. In the final version, the authours are advisable to futher integrate the rebuttal and improve the paper quality.